# O-GlcNAc modification of leucyl-tRNA synthetase 1 integrates leucine and glucose availability to regulate mTORC1 and the metabolic fate of leucine

Kibum Kim [1,2,10], Hee Chan Yoo [2,10], Byung Gyu Kim [3], Sulhee Kim [4], Yulseung Sung [2], Ina Yoon [2,5,6], Ya Chun Yu [2], Seung Joon Park [1,2], Jong Hyun Kim [7], Kyungjae Myung[3,8], Kwang Yeon Hwang[4], Sunghoon Kim [2,5,6] & Jung Min Han [1,2,9] ✉

All living organisms have the ability to sense nutrient levels to coordinate cellular metabolism. Despite the importance of nutrient-sensing pathways that detect the levels of amino acids and glucose, how the availability of these two types of nutrients is integrated is unclear. Here, we show that glucose availability regulates the central nutrient effector mTORC1 through intracellular leucine sensor leucyl-tRNA synthetase 1 (LARS1). Glucose starvation results in O-GlcNAcylation of LARS1 on residue S1042. This modification inhibits the interaction of LARS1 with RagD GTPase and reduces the affinity of LARS1 for leucine by promoting phosphorylation of its leucine-binding site by the autophagy-activating kinase ULK1, decreasing mTORC1 activity. The lack of LARS1 O-GlcNAcylation constitutively activates mTORC1, supporting its ability to sense leucine, and deregulates protein synthesis and leucine catabolism under glucose starvation. This work demonstrates that LARS1 integrates leucine and glucose availability to regulate mTORC1 and the metabolic fate of leucine.

---

[1] Interdisciplinary Program of Integrated OMICS for Biomedical Science, Graduate School, Yonsei University, Seoul 03722, South Korea. [2] Yonsei Institute of Pharmaceutical Sciences, College of Pharmacy, Yonsei University, Incheon 21983, South Korea. [3] Center for Genomic Integrity, Institute for Basic Science, Ulsan 44919, South Korea. [4] Department of Biotechnology, College of Life Sciences and Biotechnology, Korea University, Seoul 02841, South Korea.
[5] Institute for Artificial Intelligence and Biomedical Research, Medicinal Bioconvergence Research Center, Yonsei University, Incheon 21983, South Korea.
[6] College of Medicine, Gangnam Severance Hospital, Yonsei University, Seoul 06273, South Korea. [7] Department of Biochemistry, School of Medicine, Catholic University of Daegu, Daegu 42472, South Korea. [8] Department of Biomedical Engineering, Ulsan National Institute of Science and Technology, Ulsan 44919, South Korea. [9] POSTECH Biotech Center, Pohang University of Science and Technology, Pohang 37673, South Korea. [10] These authors contributed equally: Kibum Kim, Hee Chan Yoo. ✉email: jhan74@yonsei.ac.kr

Nutrient sensing, one of the most essential cellular functions, enables the coordination of changes in cellular metabolism with the environment[1]. Although understanding of the mechanism by which cells sense and adapt to nutrient availability in the surroundings is not complete, the mechanistic target of rapamycin complex 1 (mTORC1), a crucial factor for nutrient signaling, appears to be a master nutrient effector that stimulates anabolic programs, including the synthesis of proteins, nucleotides, and lipids, under conditions of sufficient nutrient availability, and suppresses catabolic programs, such as autophagy[2,3]. An extensive investigation of the regulator of Rag and Rheb GTPases revealed the integration of a variety of cues, including amino acids and glucose, to activate mTORC1[4–8].

Rag GTPases, amino acid-responsive mediators of the mTORC1 pathway[4,9], form obligate heterodimers of either RagB/RagD or RagA/RagC and are responsible for amino acid-dependent mTORC1 signaling at the lysosomal surface[4,9,10]. In response to an abundance of amino acids, these amino acids control GTP loading of Rag GTPases via guanine nucleotide exchange factor (GEF)[6] or guanosine triphosphatase-activating protein (GAP)[11,12]. The nucleotide status of Rag GTPases regulates their interaction with the regulatory-associated protein of mTOR (Raptor), a subunit of mTORC1. Raptor favorably interacts with active Rag GTPases composed of GTP-bound RagA or RagB and GDP-bound RagC or RagD, leading them to lysosomes, where they interact with Ragulator, which is localized on the lysosomal membrane[13]. Leucyl-tRNA synthetase 1 (LARS1) catalyzes the ligation of a leucine to its cognate tRNA using ATP, which is termed leucylation. LARS1 was first identified as a leucine sensor for mTORC1 due to its function as a GAP for RagD[14]. Leucine leads to the translocation of LARS1 and mTORC1 to the lysosome via LARS1-dependent GTP hydrolysis of RagD, which activates mTORC1 signaling[14–16]. Since the kinetics of mTORC1 activity are well correlated with the GTP/GDP status of RagB/RagD, the GTP hydrolysis process of RagB/RagD is considered a rate-limiting step for mTORC1 activation[4,16,17].

Activated mTORC1 augments protein synthesis by phosphorylating T389 of p70 ribosomal protein S6 kinase 1 (S6K1)[18] and T36/T47 of eukaryotic initiation factor 4E-binding protein 1 (4E-BP1)[19]. Upon its phosphorylation by mTORC1, S6K1 increases the transcription of ribosomal RNA, supporting newly assembled ribosomes, resulting in enhanced protein synthesis[18] and 4E-BP1 releases eIF4E and increases the global translation of mRNAs[20].

Glucose is also sensed through mTORC1 via several glucose sensors. The best-characterized regulator of mTORC1 activity in response to glucose is AMP-activated protein kinase (AMPK)[21]. Under glucose starvation, AMPK activates tuberous sclerosis complex 2 (TSC2)[5] and phosphorylates regulatory-associated protein of mTOR (Raptor), resulting in mTORC1 suppression[22]. Recently, aldolase has been reported as a glucose sensor. Aldolase, as a glycolytic enzyme, unoccupied by its substrate fructose-1,6-bisphosphate (FBP) under low-glucose conditions, promotes the formation of the AXIN-based AMPK activation complex[23] and suppresses mTORC1 signaling[24].

O-linked N-acetylglucosamine glycosylation (O-GlcNAcylation) is another type of glucose sensor in cell metabolism. O-GlcNAcylation, one of the most common post-translational protein modifications, utilizes UDP-GlcNAc generated via the hexosamine biosynthesis pathway and is regulated by two enzymes: O-GlcNAc transferase (OGT) transfers UDP-GlcNAc to the serine and threonine residues of proteins, while O-GlcNAcase (OGA) hydrolyzes O-GlcNAc and thus releases it from proteins[25]. Since cellular glucose is metabolized through hexosamine biosynthesis, which consumes other essential metabolites such as glutamine, acetyl-CoA, and UTP for cell growth, O-GlcNAcylation has emerged as a glucose sensor that regulate signaling, transcription, and cell fate[26,27]. Interestingly, glucose starvation increased global protein O-GlcNAcylation in cells through several mechanisms[28–30], including enhanced degradation of intracellular glycogen providing a source for UDP-GlcNAc[31]. However, in other cases, cellular O-GlcNAcylation is dependent on the level of glucose[32,33]. Although it is unclear what causes different responses upon glucose availability, this may imply the importance of the O-GlcNAcylation as a glucose sensor in mammalian cells.

Recent evidence indicates that LARS1 might regulate the use of leucine for protein synthesis or energy production[34]. Upon glucose starvation, LARS1 is phosphorylated by ULK1 at the residues essential for leucine binding, resulting in decreased leucine binding, which suppresses protein translation, conserving energy for survival. Although ULK1 activation induced by glucose starvation is a well-known stress adaptation mechanism that stimulates autophagy[35,36], little is known about how the leucine-sensing LARS1-mTORC1 axis detects intracellular glucose, the most essential nutrient for cell survival.

In this study, we show that O-GlcNAcylation locks LARS1 in a "sensing-off" state upon glucose starvation, interfering with its binding with RagD and leading to mTORC1 inactivation. Subsequently, this modification facilitates ULK1-mediated phosphorylation at the catalytic site and decreases its affinity for leucine. As a result, global protein synthesis is inhibited and therefore leucine is redirected toward catabolic metabolism to support cell survival upon glucose starvation. These results suggest that the O-GlcNAcylation of LARS1 is crucial for the response to cellular stress and provide evidence that LARS1 coordinately regulates protein synthesis and metabolism by integrating leucine and glucose availability.

## Results

**LARS1 regulates glucose-dependent mTORC1 activation.** To investigate the effect of leucine availability on glucose sensing by mTORC1, we starved cells of leucine in the presence or absence of glucose and then added back leucine and analyzed the phosphorylation of mTORC1 targets over the course of 15 minutes by immunoblotting. In the presence of glucose, leucine resupplementation induced S6K T389 and ULK1 S757 phosphorylation in a time-dependent manner, but leucine-induced phosphorylation of S6K T389 and ULK1 S757 was suppressed by glucose starvation (Fig. 1a) or glycolysis inhibitor 2-deoxy-D-glucose (2-DG) treatment (Supplementary Fig. 1a). When the cells were starved of glucose in the presence or absence of leucine and then glucose was added back, likewise, we saw that the mTORC1 targets were initially dephosphorylated but became phosphorylated over the course of 5–30 minutes after addition of glucose, whereas in the absence of leucine, addition of glucose failed to activate mTORC1 (Fig. 1b). Removal of 2-DG rescued leucine-induced phosphorylation of S6K T389 and ULK1 S757 (Supplementary Fig. 1b). These data indicate that both leucine and glucose are necessary for mTORC1 activation.

To determine whether LARS1 mediates the activation of mTORC1 by both leucine and glucose, we starved cells of leucine in the presence or absence of glucose, then added back leucine for 15 minutes and analyzed the cell lysates and the lysosomal fractions for the presence of the mTORC1 subunits mTOR and Raptor and for LARS1. We observed leucine-induced enrichment of LARS1, mTOR, and Raptor in the lysosomal fraction in glucose-supplemented cells but not glucose-starved cells (Fig. 1c) or 2-DG-treated cells (Supplementary Fig. 1c). Thus, both leucine and glucose are required for the translocation of LARS1 and mTORC1 to lysosomes. We confirmed this conclusion by

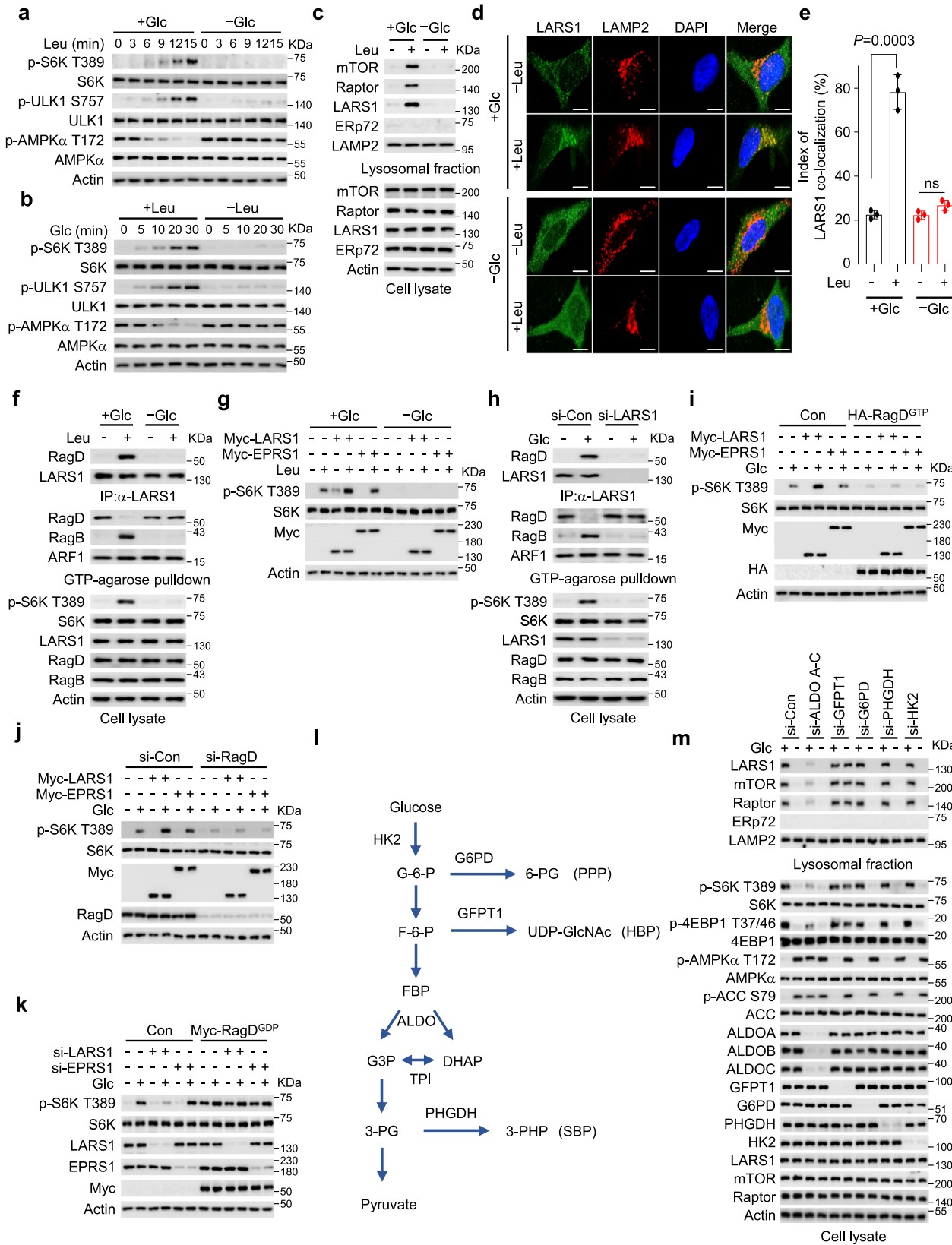

immunofluorescence microscopy of LARS1 and the lysosomal marker LAMP2 in cells. We found that the supplementation with both leucine and glucose induced the lysosomal localization of LARS1, whereas the leucine-induced lysosomal localization of LARS1 was not observed in glucose-starved cells (Fig. 1d, e).

To discover whether the interaction between LARS1 and GTP-bound RagD and the GTP hydrolysis activity of RagD require both leucine and glucose, we starved cells of leucine in the presence or absence of glucose, then added back leucine for 15 minutes. Using anti-LARS1 antibody-conjugated agarose

**Fig. 1 LARS1 regulates glucose-dependent mTORC1 activation. a** SW620 cells were starved of leucine for 1.5 h with or without 11 mM glucose, then 0.4 mM leucine was added for the indicated durations. **b** SW620 cells were starved of glucose for 4 h with or without 0.4 mM leucine, then 11 mM glucose was added for the indicated durations. **c** SW620 cells were starved of leucine for 1.5 h with or without 11 mM glucose, then 0.4 mM leucine was added for 15 min with or without 11 mM glucose. **d** HeLa cells were starved of leucine for 1.5 h with or without 25 mM glucose, then 0.8 mM leucine was added for 15 min with or without 25 mM glucose. Scale bars, 10 μm. **e** Colocalization of LARS1 and LAMP2 immunofluorescence staining of **d**. Mean±SD, $n = 3$ independent experiments. *P*-value was determined by two-tailed unpaired Student's *t* test. ns, not significant. **f** SW620 cells were starved of leucine for 1.5 h with or without 11 mM glucose, then 0.4 mM leucine was added for 15 min with or without 11 mM glucose. **g** SW620 cells were transfected with the indicated expression constructs. After 24 h, they were starved of leucine for 1.5 h with or without 11 mM glucose, then 0.4 mM leucine was added for 15 min with or without 11 mM glucose. **h**, **i**, **j**, **k** SW620 cells were transfected with the indicated expression constructs or siRNAs. After constructs or siRNAs transfection, they were starved of glucose for 4 h then 11 mM glucose was added for 30 min. **l** Summary of glucose metabolic pathways. Glucose-6-phosphate (G-6-P); fructose-6-phosphate (F-6-P); fructose-1,6-bisphosphate (FBP); dihydroxyacetone phosphate (DHAP); glyceraldehyde 3-phosphate (G3P); 3-phosphoglycerate (3-PG); 6-phosphogluconate (6-PG); UDP-N-acetylglucosamine (UDP-GlcNAc); 3-phosphohydroxypyruvate (3-PHP); Glucose-6-phosphate dehydrogenase (G6PD); Glutamine-fructose-6-phosphate transaminase 1 (GFPT1); aldolase (ALDO); triose phosphate isomerase (TPI); 3-phosphoglycerate dehydrogenase (PHGDH); pentose phosphate pathway (PPP); hexosamine biosynthesis pathway (HBP); serine biosynthesis pathway (SBP). **m** SW620 cells were transfected with the indicated siRNAs. After 48 h, the cells were starved of glucose for 4 h, then 11 mM glucose was added for 30 min. Representative data of three experiments with similar results. Source data are provided as a Source Data file.

beads or GTP-conjugated agarose beads, we purified LARS1 and GTP-bound RagD, respectively, then analyzed the bound proteins by immunoblotting with antibodies against LARS1, RagD, or RagB. We observed that the interaction between LARS1 and RagD and the GTP hydrolysis of RagD induced by leucine were observed in glucose-supplemented cells but not in glucose-starved cells (Fig. 1f) or 2-DG-treated cells (Supplementary Fig. 1d).

To clarify downstream signals to mTORC1 stimulation under glucose supplementation, we monitored the contribution of LARS1 on leucine-induced mTORC1 activity with or without glucose. Overexpression of LARS1, but not glutamyl-prolyl-tRNA synthetase 1 (EPRS1), increased S6K T389 phosphorylation in glucose-supplemented cells even without leucine (Fig. 1g) but it had no effect in glucose-starved cells (Fig. 1g). Depletion of LARS1 by siRNA-mediated gene silencing, by contrast, suppressed glucose- or 2-DG removal-induced GTP hydrolysis of RagD GTPase and S6K T389 phosphorylation (Fig. 1h and Supplementary Fig. 1e). These data indicate that LARS1 mediates mTORC1 activation induced not only by leucine but also by glucose.

To confirm the role of the RagD GTPase in glucose signaling downstream of LARS1, we overexpressed an inactive HA-tagged RagD GTPase mutant (Q121L; HA-RagD$^{GTP}$) in control cells and cells that were overexpressing Myc-LARS1, starved the cells of glucose for 4 h, and then added it back for 30 min. Overexpression of HA-RagD$^{GTP}$ or RagD siRNA transfection abolished the increase in glucose-induced S6K T389 phosphorylation induced by LARS1 overexpression (Fig. 1i, j and Supplementary Fig. 1f, g). Conversely, constitutively active Myc-tagged RagD GTPase mutant (S77L; Myc-Rag$^{GDP}$) overexpression rescued LARS1 downregulation-induced suppression of S6K T389, regardless of the presence or absence of glucose (Fig. 1k) or 2-DG treatment (Supplementary Fig. 1h). These results indicate that LARS1 plays a specific role in glucose-induced mTORC1 activation.

To investigate the roles of known glucose sensors upstream of LARS1 in mTORC1 activation, we depleted each of five key enzymes involved in glucose metabolism in cells: aldolases (ALDO A-C), glutamine-fructose-6-phosphate transaminase 1 (GFPT1), glucose-6-phosphate dehydrogenase (G6PD), 3-phosphoglycerate dehydrogenase (PHGDH) and hexokinase 2 (HK2), a negative regulator of mTORC1 in the absence of glucose[37] (Fig. 1l). Among them, the knockdown of aldolases decreased lysosomal translocation of the LARS1-mTORC1 axis and S6K T389 phosphorylation even in the presence of glucose (Fig. 1m). Furthermore, downregulation of GFPT1, a key enzyme involved in UDP-GlcNAc synthesis in the hexosamine biosynthetic pathway[38], increased

lysosomal translocation of the LARS1-mTORC1 axis and S6K T389 phosphorylation despite glucose starvation conditions (Fig. 1m). Downregulation of G6PD, PHGDH, or HK2 had little effect on glucose-induced mTORC1 activation (Fig. 1m). These data suggest that signaling through LARS1 to control mTORC1 is regulated by aldolases and GFPT1 and that the hexosamine biosynthesis pathway might be newly associated with mTORC1 signaling through LARS1.

## LARS1 is O-GlcNAcylated at S1042 by OGT1 under glucose starvation.
The hexosamine biosynthesis pathway uses glucose, acetyl-CoA, glutamine, and the nucleotide UTP to generate UDP-GlcNAc, a substrate for the post-translational modification of Ser and Thr residues in many proteins with O-linked N-acetyl glucosamine (O-GlcNAcylation). This modification controls protein functions by modulating protein–protein interactions, altering protein structure and stability[25]. We therefore examined whether LARS1, RagB or RagD are O-GlcNAcylated in response to glucose starvation by isolating the proteins from cell lysates on succinylated wheat germ agglutinin (sWGA)-conjugated agarose beads, which bind directly to the O-GlcNAc group, and then immunoblotting the bound proteins. Glucose starvation induced the O-GlcNAcylation of LARS1 irrespective of the presence or absence of leucine whereas neither RagB nor RagD were modified, indicating the specificity of O-GlcNAcylation (Fig. 2a). Consistent with a previous report, ATG4B was O-GlcNAcylated by glucose starvation[39] (Fig. 2a). This O-GlcNAcylation of LARS1 was seen also in several other cell lines (Supplementary Fig. 2a).

Many types of stress, including glucose starvation, rapidly and substantially increase O-GlcNAcylation of a variety of proteins through increased activity of OGT1[28,40]. This is paradoxical since glucose starvation might reduce intracellular glucose, a source of UDP-GlcNAc for protein O-GlcNAcylation[31]. To investigate whether the effect of intracellular UDP-GlcNAc levels on LARS1 O-GlcNAcylation, we monitored the O-GlcNAcylation levels under UDP-GlcNAc supplementation in cells with either depletion of GFPT1 or OGT1. Glucose starvation increased the global O-GlcNAcylation of proteins, including O-GlcNAcylation of LARS1 (Fig. 2b), implying that the intracellular levels of UDP-GlcNAc are sufficient for O-GlcNAcylation of proteins even under glucose starvation conditions. Conversely, in the depletion of GFPT1 or OGT1, glucose starvation did not increase the O-GlcNAcylation of LARS1 (Fig. 2b). Notably, UDP-GlcNAc supplementation only increased the O-GlcNAcylation of LARS1 under the GFTP1 depletion, but not under the OGT1 knockdown (Fig. 2b). These results suggest that the level of UDP-GlcNAc

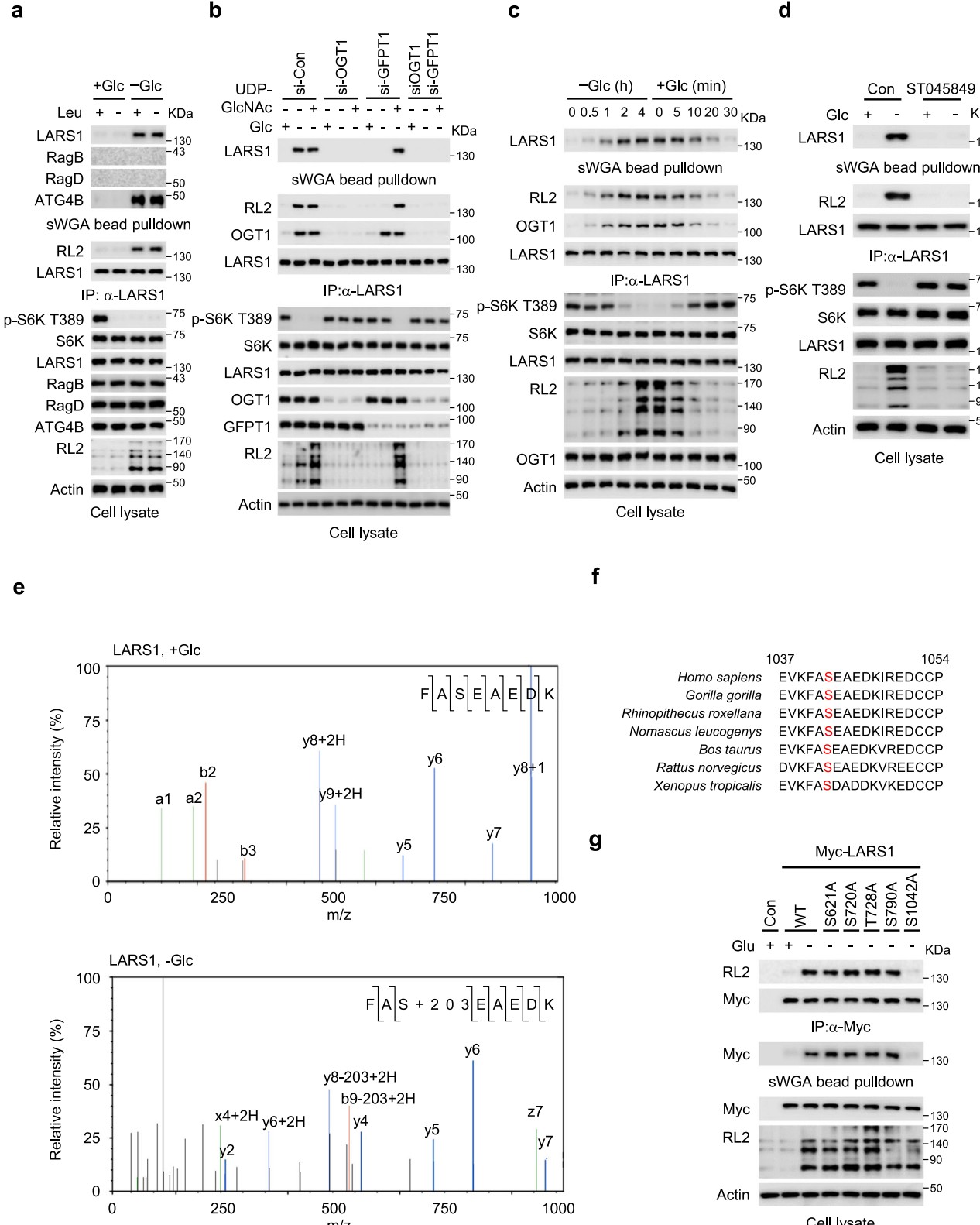

sustained by GFPT1, and the OGT1 activity are essential for the O-GlcNAcylation of LARS1 in glucose starvation.

To investigate the role of OGT1 in the O-GlcNAcylation of LARS1, we starved cells of glucose for up to four hours then added glucose for up to 30 min, immunoprecipitated and immunoblotted for LARS1, and OGT1 over the course of the experiment. In the absence of glucose, LARS1 progressively

interacted with OGT1 and was O-GlcNAcylated over time whereas, when glucose was added the interaction progressively decreased (Fig. 2c). At the same time, the level of S6K T389 phosphorylation progressively decreased in the absence of glucose and increased again when glucose was added (Fig. 2c), suggesting O-GlcNAcylation of LARS1 might negatively regulate mTORC1. Chemical inhibition (with ST045849) or downregulation of

**Fig. 2 LARS1 is *O*-GlcNAcylated at S1042 by OGT1 under glucose starvation. a** SW620 cells were incubated with or without 11 mM glucose for 4 h. **b** SW620 cells were transfected with the indicated siRNAs. After 48 h, the cells were starved of glucose for 4 h, then permeabilized with streptolysin O for 5 min, and 200 μM of UDP-GlcNAc, as indicated, was added for 30 min. **c** SW620 cells were starved of glucose and supplemented with 11 mM glucose for the indicated durations. **d** SW620 cells were incubated with vehicle or 20 μM ST045849 for 24 h and starved of glucose for 4 h. **a–d** Each cell lysate was precipitated with succinylated wheat germ agglutinin (sWGA)-conjugated agarose beads or anti-LARS1 antibody-conjugated agarose beads and then immunoblotted with the indicated antibodies. **e** The site of LARS1 *O*-GlcNAcylation was mapped using mass spectrometry. **f** LARS1 protein sequences, including S1042, the *O*-GlcNAcylation site, among species were aligned using BLAST. Red alphabet indicates a conserved serine residue. **g** SW620 cells were transfected with the indicated LARS1 constructs. After 24 h, the cells were starved of glucose for 4 h. Each sample was precipitated with sWGA-conjugated agarose beads or anti-LARS1 antibody-conjugated agarose beads and analyzed by immunoblotting with the indicated antibodies. Representative data of three experiments with similar results. Source data are provided as a Source Data file.

OGT1 suppressed LARS1 *O*-GlcNAcylation induced by glucose starvation (Fig. 2d and Supplementary Fig. 2b, c).

We showed above that aldolase downregulation decreased the translocation of the LARS1 to lysosomes and decreased S6K T389 phosphorylation (Fig. 1m). To investigate whether aldolase is also involved in the *O*-GlcNAcylation of LARS1, we silenced the aldolase genes in cells. Depletion of aldolases had no effect on LARS1 *O*-GlcNAcylation induced by glucose starvation (Supplementary Fig. 2d), suggesting that aldolases are not correlated with LARS1 *O*-GlcNAcylation.

To identify the site(s) on LARS1 that are *O*-GlcNAcylated, we overexpressed a Strep-tagged LARS1 protein in SW620 cells, purified the protein from glucose-starved cells by binding first to streptavidin beads and then to sWGA-conjugated beads, and subjected the purified protein to proteolytic digestion and mass spectrometry. In these conditions of glucose starvation, LARS1 was specifically *O*-GlcNAcylated at a single site, S1042 (Fig. 2e). There were no other posttranslational modifications at S1042, regardless of the presence or absence of glucose. Residue S1042 is conserved in LARS1 from various species (Fig. 2f). To confirm that S1042 is the major site in LARS1 that is modified by OGT1-catalyzed *O*-GlcNAcylation under glucose starvation, we mutated four Ser and one Thr residue in LARS1 to Ala and overexpressed the mutated Myc-tagged LARS1 in cells. All of the Myc-tagged mutant proteins except the S1042A mutant bound to sWGA-conjugated agarose, indicating that S1042 is the major *O*-GlcNAcylated residue under glucose starvation (Fig. 2g).

**The *O*-GlcNAcylation of LARS1 suppresses leucine-induced mTORC1 activation.** To determine the effect of LARS1 *O*-GlcNAcylation on mTORC1 activation, we analyzed the kinetics of S6K T389 phosphorylation over the course of 15 minutes after addition of leucine to cells overexpressing the wild-type (WT) LARS1 or the S1042A mutant in control cells. Leucine-induced S6K T389 phosphorylation was faster in control cells expressing the S1042A mutant than in those expressing the WT in the presence of glucose. Moreover, leucine-induced S6K T389 phosphorylation was more pronounced in control cells expressing the S1042A mutant than in those expressing the WT in the absence of glucose (Fig. 3a, b). The S1042A mutant promoted leucine-induced translocation of mTORC1 to lysosomes and S6K T389 phosphorylation even in the absence of glucose, whereas WT LARS1 did not (Fig. 3c). Importantly, LARS1 S1042A mutation impaired LARS1's GAP activity for RagD, resulting in leucine-induced GTP hydrolysis of RagD even in glucose starvation conditions (Fig. 3d), displaying that the lack of LARS1 *O*-GlcNAcylation on S1042A causes constitutive activation of mTORC1 despite glucose starvation.

To demonstrate the importance of *O*-GlcNAcylation for the regulation of endogenous LARS1 activation of mTORC1, we used CRISPR/Cas9 to generate LARS1 S1042A knock-in cells (Fig. 3e, f). We starved these knock-in cells and WT cells of glucose for 4 h, added glucose for 30 min and then analyzed the cell lysates for S6K

phosphorylation on T389, immunoprecipitated and immunoblotted for endogenous LARS1 (Fig. 3g). LARS1 S1042A knock-in cells displayed enhanced S6K T389 phosphorylation (Fig. 3g) and lysosomal enrichment of mTOR, Raptor, and LARS1 proteins upon glucose starvation when compared to the control, WT cells (Fig. 3h). These findings are all consistent with the notion that LARS1 *O*-GlcNAcylation negatively regulates mTORC1 activation.

The stress-induced protein Sestrin is reported to be a leucine sensor[41]. To investigate the involvement of Sestrin in LARS1 S1042A mutant induced activation of mTORC1, we silenced the genes encoding Sestrin 1 and Sestrin 2 and assayed the effect on LARS1 S1042A mutant-induced S6K T389 phosphorylation and on RagD and RagB binding to GTP-agarose as measures of signaling to mTORC1. Sestrin1/2 knockdown affected only the GTP exchange of RagB and had no effect on S1042A LARS1-induced RagD GTP hydrolysis (Fig. 3i), consistent with our previous report indicating that LARS1 and Sestrin 2 have distinct roles in regulating the Rag GTPase cycle[16]. These data indicate that the *O*-GlcNAcylation of LARS1 at residue S1042 mediated by OGT1 in glucose-starved cells is important for leucine signaling to mTORC1.

In addition to mTORC1 signaling, we investigated whether the *O*-GlcNAcylation of LARS1 affects the leucylation activity of LARS1. To end this, we expressed LARS1 WT and S1042A proteins in *E.coli* with or without OGT1, isolated, and measured the stoichiometry of *O*-GlcNAcylated LARS1 using GalTY289L labeling. We observed that about 95% of WT LARS1 was *O*-GlcNAcylated, and the S1042A mutant was not *O*-GlcNAcylated at all when LARS1 was co-expressed with OGT1 in *E.coli*. (Supplementary Fig. 3a). However, there was no obvious difference of leucylation activity between WT and S1042A mutant, either OGT1 expression or not, indicating that *O*-GlcNAcylation itself has no effect on the catalytic activity of LARS1 (Supplementary Fig. 3b).

**O-GlcNAcylation of LARS1 S1042 inhibits the interaction of LARS1 with RagD.** Human cytosolic LARS1 comprises five domains: a catalytic domain (CD; residues 1–255 and 515–763); a CP1/editing domain (CP1; residues 256–514); a tRNA anticodon-binding domain (ABD; residues 764–892); a vertebrate C-terminal domain (VC; residues 893–1062), and a C-terminal UNE-L domain (residues 1063–1176)[42]. From recently published crystal structures of 'sensing-off' (PDB entry 6KR7) and 'sensing-on' (PDB entry 6KQY) forms of LARS1 in complex with leucine and ATP[43], we can see that the VC domain of LARS1 contains the S1042 *O*-GlcNAcylation site (Supplementary Fig. 4a) and the site that binds the RagD GTPase[14]. To obtain insights into how *O*-GlcNAcylation of LARS1 on S1042 might affect the conformation of the VC domain, we compared the published structures of the sensing-off (Fig. 4a) and sensing-on (Fig. 4b) forms of the LARS1 VC domain (residues 948–1015) with the same domain after *O*-GlcNAcylation as determined by modelling (Fig. 4c; see Fig. 4d for a super-position of the three structures).

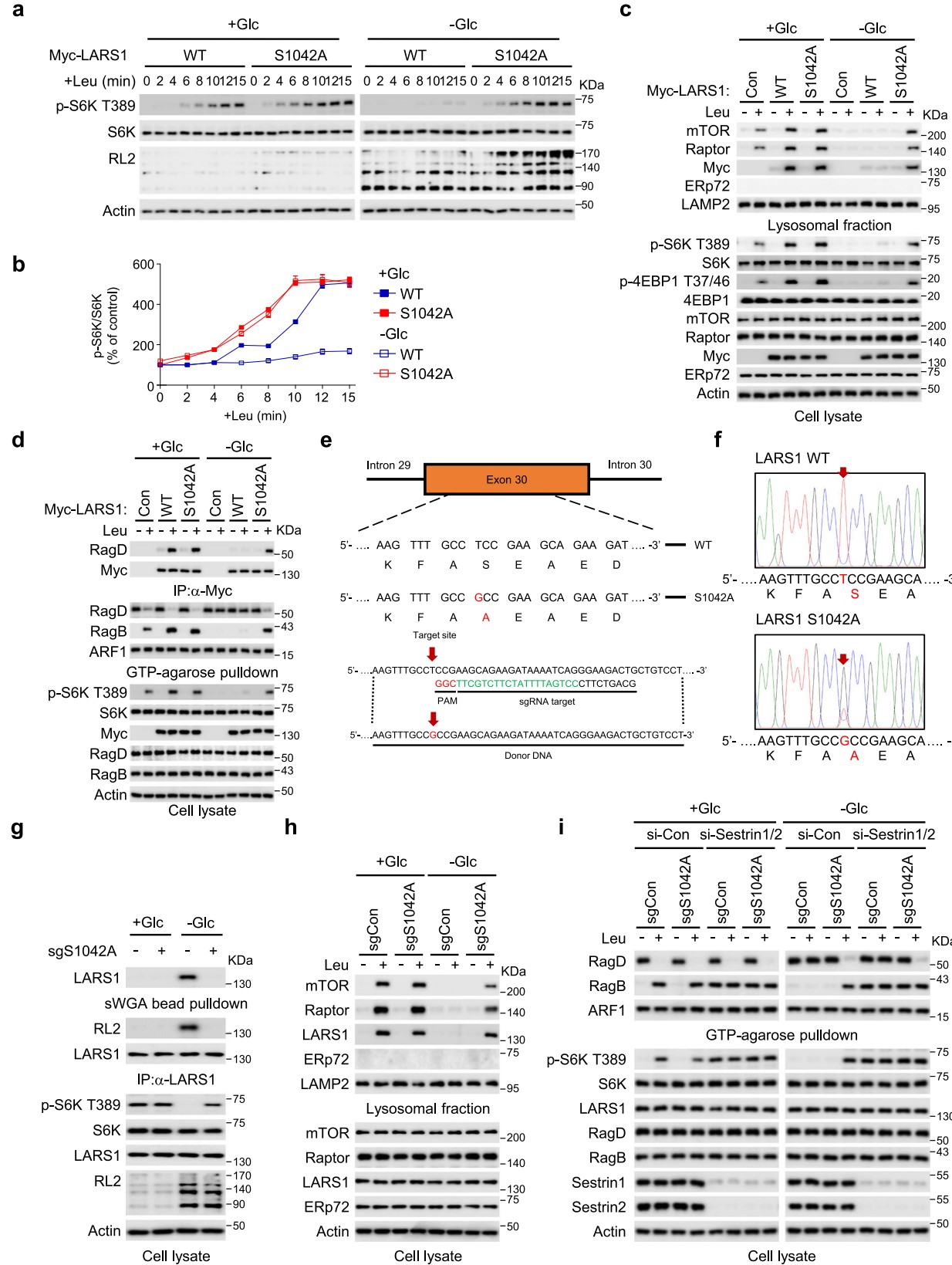

The root-mean-square deviation (RMSD) between *O*-GlcNAcylated S1042 LARS1 and the sensing-off structure was 1.903 Å, the RMSD between the sensing-on and sensing-off structures was 1.380 Å, and the RMSD between *O*-GlcNAcylated S1042 LARS1 and the sensing-on structure was 2.251 Å (Fig. 4d). Overall, the folds of the three structures were similar, but a detailed analysis of

their surfaces, especially the area including residues H958, E960, and K970, which are involved in binding RagD[14,43], were remarkably different (Fig. 4d).

We verified the local conformational change in the VC domain of LARS1 by using near-UV circular dichroism spectra analysis. The spectrum of the S1042A mutant, which does not undergo *O*-

**Fig. 3 The *O*-GlcNAcylation of LARS1 suppresses leucine-induced mTORC1 activation. a** SW620 cells were transfected with the indicated LARS1 constructs. After 24 h, the cells were starved of leucine for 1.5 h and supplemented with 0.4 mM leucine for the indicated durations. Each cell lysate was immunoblotted with the indicated antibodies. **b** Quantification of S6K T389 phosphorylation of **a**. The value for the control sample at 0 min was set to 100% (mean ± SEM, $n = 3$ independent experiments). **c, d** SW620 cells were transfected with the indicated LARS1 constructs. After 24 h, the cells were starved of 0.4 mM leucine for 1.5 h and supplemented with 0.4 mM leucine for 15 min with or without 11 mM glucose. **c** Each cell lysate was used for lysosomal fractionation and immunoblotted with the indicated antibodies. **d** Each cell lysate was precipitated with anti-LARS1 antibody-conjugated agarose beads or GTP-conjugated agarose beads and analyzed by immunoblotting with the indicated antibodies. **e** A schematic of the LARS1 genomic location and the selected sgRNA targeting site. **f** Sequencing result of targeted genomic regions of LARS1. Red arrows point to overlapping peaks. **g** SW620 control or S1042A knock-in cells were starved of glucose for 4 h and supplemented with 11 mM glucose for 30 min. Each cell lysate was precipitated with sWGA-conjugated agarose beads or anti-LARS1 antibody-conjugated agarose beads and then immunoblotted with the indicated antibodies. **h** SW620 control or S1042A knock-in cells were starved of 0.4 mM leucine for 1.5 h and supplemented with 0.4 mM leucine for 15 min with or without 11 mM glucose. Each cell lysate was used for lysosomal fractionation and immunoblotted with the indicated antibodies. **i** SW620 control or S1042A knock-in cells were transfected with the indicated siRNAs. After 48 h, cells were starved of 0.4 mM leucine for 1.5 h and supplemented with 0.4 mM leucine for 15 min with or without 11 mM glucose. Each cell lysate was precipitated with anti-LARS1 antibody-conjugated agarose beads or GTP-conjugated agarose beads and analyzed by immunoblotting with the indicated antibodies. Representative data of three experiments with similar results. Source data are provided as a Source Data file.

GlcNAcylation, was similar to that of WT LARS1 (Fig. 4e). When the proteins were purified from cells overexpressing OGT1, by contrast, the spectra of the S1042A mutant and WT LARS1, which we presume was *O*-GlcNAcylated, were quite different in the region of 260–290 nm (Fig. 4e), indicating a significantly different local tertiary structure in the *O*-GlcNAcylated protein. This region of the near-UV circular dichroism spectrum has been assigned to tyrosine residues[44]. The VC domain of LARS1 contains 6 tyrosine residues. We identified the location of these tyrosine residues in the VC domain structure and analyzed their mobility using in silico-modeling (Supplementary Fig. 4a, b). *O*-GlcNAcylation of S1042 induced a local change in the tertiary structure around these residues in the VC domain and distorted an α-helix that includes the RagD-binding site by 17.2 degrees compared to its position in the sensing-on structure (Supplementary Fig. 4a). These tyrosine residues had high fluctuation scores in a structural flexibility profile of the VC domain (Supplementary Fig. 4b), and they are highly conserved in primates (Supplementary Fig. 4c). We investigated this conformational change in *O*-GlcNAcylated LARS1 further by using brief proteolysis of His-tagged WT and S1042A mutant LARS1 proteins expressed in *E.coli* either with WT OGT1 or with an inactive OGT1 mutant N567K. Proteinase K cleavage of WT LARS1 was faster when it was co-expressed with WT OGT1 than when it was co-expressed with the N567K mutant whereas no difference was seen when the S1042A mutant of LARS1 was co-expressed with WT or N567K mutant OGT1 (Fig. 4f). This data provides further evidence that the *O*-GlcNAcylation of S1042 is critical for the structural change in LARS1.

To determine whether the structural change in LARS1 induced by *O*-GlcNAcylation affects its binding to RagD and GTP hydrolysis of RagD, we expressed His-tagged WT and S1042A mutant LARS1 proteins in *E.coli* with or without WT or N567K mutant OGT1, isolated the His-tagged proteins on Ni-NTA beads and incubated them with purified Myc-tagged RagD or HA-tagged RagC in the presence or absence of leucine and ATP. The control, unmodified WT LARS1 bound RagD but not RagC and induced GTP hydrolysis by RagD (seen as no Myc-RagD binding to GTP-agarose) when both ATP and leucine were present (lane 2) (Fig. 4g). *O*-GlcNAcylated WT LARS1 did not bind RagD and did not induce the GTP hydrolysis by RagD (seen as Myc-RagD binding to GTP-agarose; lane 4) (Fig. 4g). By contrast, the S1042A mutant of LARS1 bound RagD (lanes 14, 16 and 18), inducing the GTP hydrolysis by RagD when both ATP and leucine were present (Fig. 4g).

To investigate the effect of LARS1 *O*-GlcNAcylation on mTORC1 signaling, we transfected cells with constructs expressing Myc-tagged LARS1 WT or the S1042A mutant, or a S621A

mutant, starved the cells of glucose for 4 h and then added back glucose for 30 min. We monitored the LARS1–RagD interaction by immunoblotting the proteins bound to GTP-agarose and anti-Myc antibody-conjugated agarose beads and monitored mTORC1 activation by the presence of S6K phosphorylated on T389 in the cell lysates (Fig. 4h). Upon either glucose starvation, the LARS1 WT and the S621A mutant, both of which would be expected to by *O*-GlcNAcylated, did not bind RagD and did not result in S6K T389 phosphorylation, whereas the S1042A mutant, which is not *O*-GlcNAcylated, did bind RagD and resulted in elevated S6K T389 phosphorylation (Fig. 4h, i). In addition, exogenous inactive GTP-loaded RagD (Q121L) suppressed the enhanced mTORC1 signaling induced by the S1042A mutation (Fig. 4j, k). These data indicate that LARS1 *O*-GlcNAcylation on S1042 prevents the interaction of LARS1 with RagD and regulates mTORC1 signaling upstream of Rag GTPase in response to glucose availability.

**The *O*-GlcNAcylation of LARS1 regulates ULK1-mediated LARS1 phosphorylation.** In the absence of glucose, ULK1 phosphorylates LARS1 on S720, a residue crucial for binding leucine, and S391, thus inhibiting leucine-induced mTORC1 activation[34]. To determine whether *O*-GlcNAcylation on S1042 affects phosphorylation of LARS1 on S720, we monitored the phosphorylation and *O*-GlcNAcylation of LARS1 over time in cells in the presence and absence of glucose by immunoprecipitation and immunoblotting. Over the course of 4 hours without glucose, the *O*-GlcNAcylated form and the S720 phosphorylated form of LARS1 gradually appeared then rapidly disappeared upon addition of glucose (Fig. 5a, b). Consistent with this, mTORC1 activity, as measured by S6K T389 phosphorylation in cell lysates, gradually decreased in the absence of glucose and rapidly reappeared upon addition of glucose (Fig. 5a). Quantification of this data showed that LARS1 *O*-GlcNAcylation was almost simultaneous with the LARS1 phosphorylation during the course of 4 h glucose starvation (Fig. 5b), suggesting that *O*-GlcNAcylation may be related to the phosphorylation of LARS1 S720. To investigate the possible dependence of LARS1 S720 phosphorylation on *O*-GlcNAcylation, we silenced the genes encoding OGT1 or GFPT1 to deplete the enzyme or its UDP-GlcNAc substrate, respectively, in cells. Glucose starvation induced both *O*-GlcNAcylation and S720 phosphorylation, whereas depletion of OGT1 or GFPT1 decreased the *O*-GlcNAcylation and markedly diminished phosphorylation at S720 (Fig. 5c). indicating that *O*-GlcNAcylation may be a prerequisite for the phosphorylation of LARS1 S720.

Glucose deprivation elevates the intracellular AMP/ATP ratio and activates the AMPK signaling pathway[21]. Since the target of

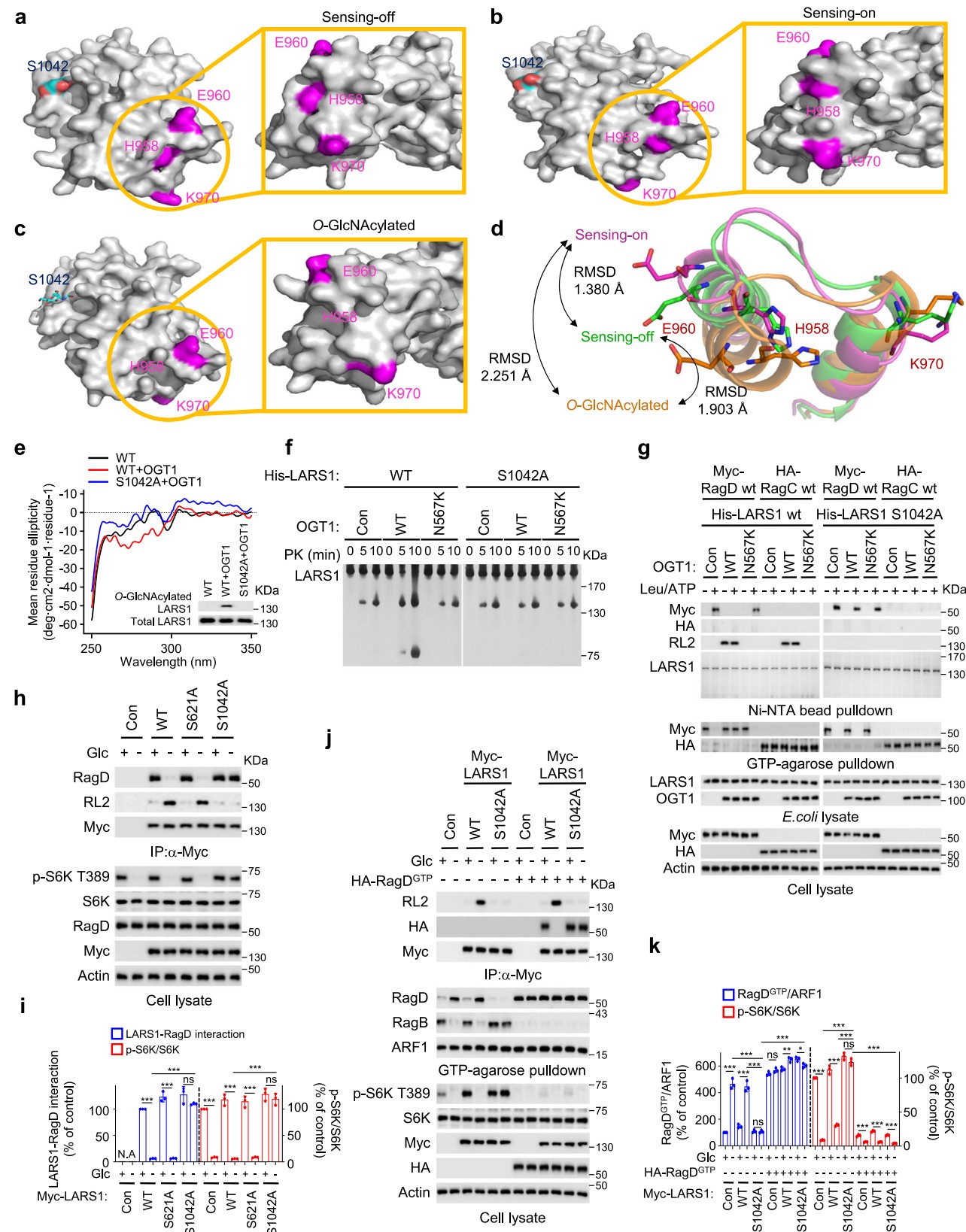

AMPK[35,36], ULK1 phosphorylates LARS1 on S720[34], we tested whether AMPK/ULK1 pathway is involved in the process of O-GlcNAcylation and phosphorylation of LARS1. Downregulation of AMPK and ULK1/2 suppressed LARS1 S720 phosphorylation but not its O-GlcNAcylation induced by glucose starvation (Fig. 5d). Although FBP and DHAP, which are known for

metabolites for AMPK inhibition[23], decreased LARS1 S720 phosphorylation induced by glucose starvation, neither of the treatments affected LARS1 O-GlcNAcylation. Consistent with these findings, the AMPK inhibitor Compound C or the ULK1 inhibitor SBI-0206965 decreased the S720 phosphorylation of LARS1, whereas they did not alter the level of LARS1 O-

**Fig. 4 O-GlcNAcylation of LARS1 S1042 inhibits the interaction of LARS1 with RagD. a–c** A surface representation of the RBD-VC domain structure of "sensing-off" LARS1 (PDB: 6KR7), "sensing-on" LARS1 (PDB: 6KQY), and O-GlcNAcylated LARS1 (modeled by Vienna PTM) were colored gray. S1042 is colored cyan, and H958, E960, and K970 are colored magenta. **d** Superimposition of the RBD-VC domain structures of "sensing-off" LARS1, "sensing-on" LARS1, and O-GlcNAcylated LARS1. "Sensing-off" LARS1, "sensing-on" LARS1, and O-GlcNAcylated LARS1 are represented as green, pink, and orange cartoon models, respectively. **e** Near-UV CD spectra of WT LARS1 and S1042A mutant LARS1 with or without OGT1. **f** His-tagged WT and S1042A mutant LARS1 were purified from E. coli with or without WT or N567K OGT1 expression. Each LARS1 protein was incubated with proteinase K for the indicated duration. **g** His-tagged WT and S1042A mutant LARS1 were purified with Ni-NTA beads from E. coli with or without WT or N567K OGT1 expression. Each LARS1 protein sample was incubated with the WT RagD or WT RagC in the presence or absence of 200 μM ATP/2 mM leucine. For in vitro GTPase assay, His-tagged WT or S1042A LARS1 proteins purified from E. coli expressing WT or N567K OGT1 were incubated with WT RagD or WT RagC proteins in the presence or absence of 200 μM ATP/2 mM leucine. After 1 h, samples were precipitated with GTP-conjugated beads. **h, j** SW620 cells were transfected with the indicated LARS1 constructs. After 24 h, the cells were starved of glucose for 4 h and supplemented with 11 mM glucose for 30 min. **i** The LARS1-RagD interaction and S6K T389 phosphorylation of **h** were quantified and are indicated as the percentage of the control without glucose or OGT1 (mean ± SEM, n = 3 independent experiments). **k** GTP-bound RagD and S6K T389 phosphorylation of **j** were quantified and are indicated as the percentage of the control without glucose or GTP-bound RagD (Q121L) (mean ± SEM, n = 3 independent experiments). P-value was determined by two-tailed unpaired Student's t test. *, P < 0.05; **, P < 0.01; ***, P < 0.001; ns, not significant. Representative data of three experiments with similar results. Source data are provided as a Source Data file.

GlcNAcylation (Fig. 5e). The ULK1 activator LYN1604 rescued the S720 phosphorylation of LARS1 in AMPK-downregulated cells, but the AMPK activator AICAR could not rescue the S720 phosphorylation of LARS1 in ULK1-downregulated cells (Fig. 5e), indicating that AMPK/ULK1 pathway controls the S720 phosphorylation of LARS1 under glucose starvation.

Although it was reported that AMPK activation decreased the GFPT1 activity lowering O-GlcNAc levels in cells[45,46], we observed that glucose starvation increased the global O-GlcNAcylation level in both AMPK γ1 WT and knock-out cells (Supplementary Fig. 5a). Furthermore, there was little effect of LARS1 O-GlcNAcylation between AMPKγ1 knock-out cells and WT control cells, whereas the LARS1 S720 phosphorylation was considerably reduced in AMPKγ1 knock-out cells (Supplementary Fig. 5a), suggesting that O-GlcNAcylation and phosphorylation of LARS1 are separately regulated under glucose starvation.

To investigate further whether the phosphorylation of LARS1 on S720 depends on its O-GlcNAcylation on S1042, we expressed WT, phosphorylation-defective (S720A), phosphomimetic (S720D), and O-GlcNAcylation-defective (S1042A) mutants of LARS1 in glucose-starved cells (Fig. 5f). In cells expressing WT LARS1, glucose starvation induced both the O-GlcNAcylation and S720 phosphorylation of LARS1 (Fig. 5f). Neither the S720A nor the S720D mutant affected the O-GlcNAcylation of LARS1 induced by glucose deprivation (Fig. 5f). However, in cells expressing the S1042A mutant, no phosphorylation of LARS1 S720 was seen (Fig. 5f). These data indicate that O-GlcNAcylation of LARS1 occurs before its phosphorylation on S720.

To test whether the O-GlcNAcylation of LARS1 regulates its interaction with ULK1 for its S720 phosphorylation, we depleted cells of OGT1 and starved them of glucose for up to 4 h before immunoprecipitating LARS1 and immunoblotting for LARS1 S720 phosphorylation and co-immunoprecipitation with ULK1. In the control cells, LARS1 S720 phosphorylation and binding of LARS1 to ULK1 increased steadily over the 4 h of glucose starvation whereas in cells depleted of OGT1, there was no LARS1–ULK1 binding and no ULK1-mediated LARS1 S720 phosphorylation (Fig. 5g). These data suggest that the O-GlcNAcylation of LARS1 regulates its phosphorylation via interaction with ULK1.

To investigate further the role of O-GlcNAcylation in the binding of LARS1 to ULK1, we purified recombinant His-tagged LARS1 WT or S1042A proteins from E.coli that were co-expressing, or not co-expressing OGT1, incubated the purified LARS1 proteins with the lysates of cells expressing Flag-ULK1 and assayed LARS1 binding to ULK1 by Ni-NTA-agarose and immunoblotting. Only the WT LARS1 purified from E. coli

expressing OGT1 – the only form that was O-GlcNAcylated – interacted with ULK1 (Supplementary Fig. 5b), indicating that O-GlcNAcylation is required for ULK1 recruitment to LARS1 for its phosphorylation at S720. To determine the peptide region of LARS1 that is involved in the interaction with ULK1, we prepared different deletion mutants of LARS1, incubated them with Flag-tagged ULK1 and tested which mutant affected the interaction with ULK1. While the O-GlcNAcylated peptides spanning 1-1176 (full length) and 1-1062 of LARS1 bound to ULK1, the peptide spanning 1-892 lost its binding capability (Supplementary Fig. 5c), implying the O-GlcNAcylated S1042 of the VC domain (residues 893-1062) of LARS1 is also required for the interaction with ULK1.

To determine whether the interaction between ULK1 and O-GlcNAcylated LARS1 depends on glucose availability, we expressed Strep-tagged LARS1 WT and S1042A mutant in cells, starved them of glucose for 4 h then added glucose for 30 min before precipitating LARS1 and its bound proteins with Strep-beads and immunoblotting for ULK1 and RagD. In the absence of glucose, LARS1 WT bound to ULK1 but not to RagD, whereas the S1042A mutant bound very little to ULK1 but did bind RagD (Supplementary Fig. 5d), suggesting that O-GlcNAcylation is required for the interaction between LARS1 and ULK1 upon glucose starvation.

To investigate the role of O-GlcNAcylation in mTORC1 activity upon glucose starvation, we expressed WT, the S1042A mutant or an S1042A, S720D (phosphomimetic) double mutant LARS1 in cells and assayed mTORC1 activation as S6K T389 phosphorylation when the cells were starved of glucose for 4 h (Fig. 5h). In these cells, S6K was phosphorylated in the absence of glucose only when the S1042A mutant was overexpressed (Fig. 5h), suggesting that the lack of O-GlcNAcylated LARS1 constitutively activates mTORC1. When the cells were depleted of ULK1, S6K was phosphorylated when the WT or the S1042A mutant were overexpressed but not when the S1042A, S720D double mutant was overexpressed (Fig. 5h), indicating the dominant role of LARS1 S720 phosphorylation for mTORC1 suppression. Also, expression of an inactive GTP-loaded RagD mutant (Q121L) in cells depleted of ULK1 abolished the S6K phosphorylation due to overexpression of WT or S1042A mutant LARS1 (Fig. 5h), indicating that O-GlcNAcylation of LARS1 controls mTORC1 activity in RagD-dependent manner.

To demonstrate the importance of O-GlcNAcylation for the regulation of endogenous LARS1 activation of mTORC1, we used LARS1 S1042A knock-in cells. We starved these knock-in cells and WT cells of glucose for 4 h, added glucose for 30 min and then analyzed the cell lysates for S6K phosphorylation on T389, immunoprecipitated LARS1 and analyzed the bound proteins by

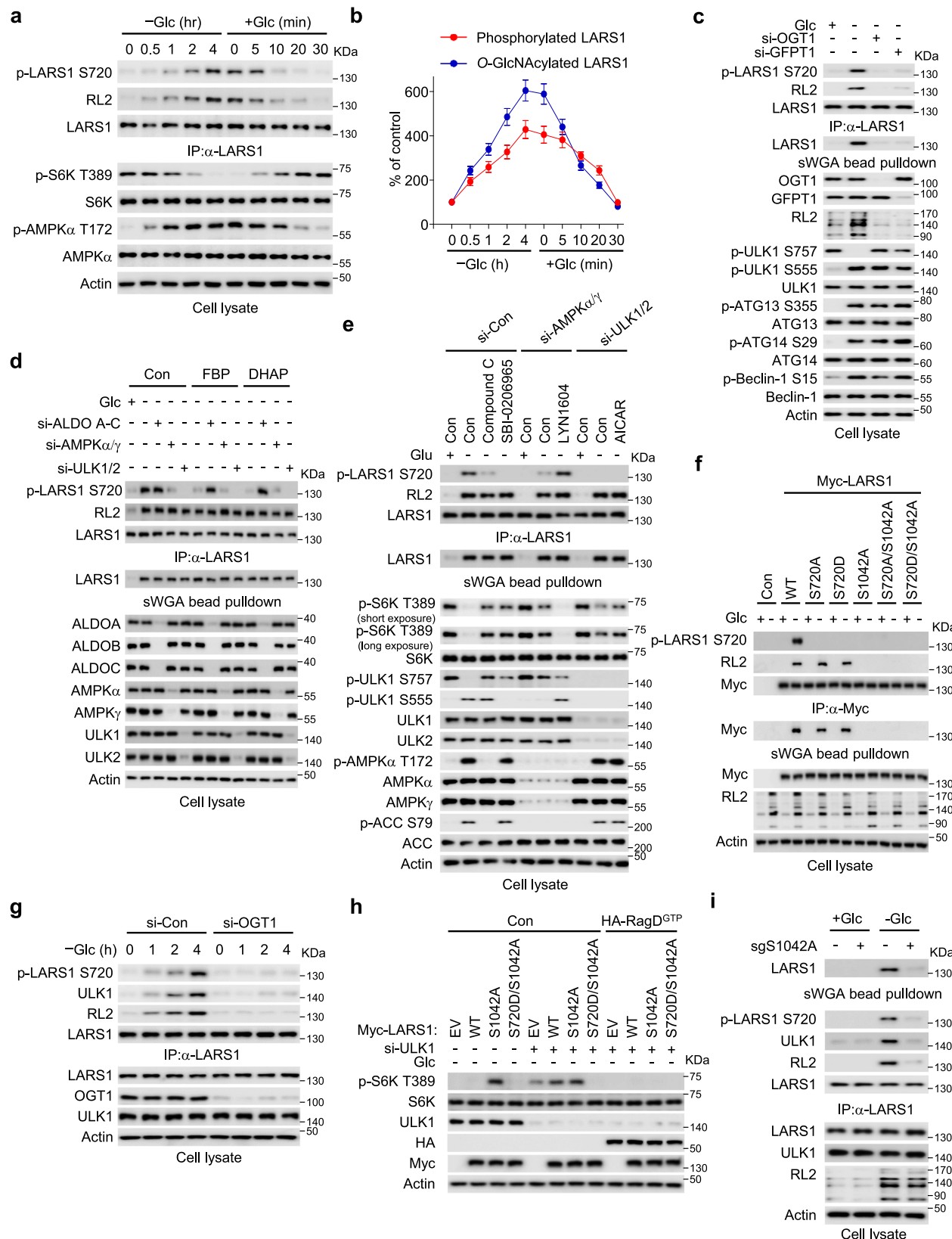

immunoblotting, and bound the O-GlcNAcylated proteins on sWGA-agarose and immunoblotted for endogenous LARS1 (Fig. 5i). In the absence of glucose, LARS1 S1042A in the knock-in cells bound much less well to sWGA and to ULK1 than did the endogenous, WT LARS1 in control cells, and it was much less phosphorylated on S720 (Fig. 5i). Moreover, the knock-in

cells displayed enhanced S6K T389 phosphorylation upon glucose starvation when compared to the control, WT cells (Fig. 3e). These data indicate that O-GlcNAcylation of endogenous LARS1 at S1042 regulates its own phosphorylation by ULK1 and acts as an upstream regulator of Rag-dependent mTORC1 activation in the absence of glucose.

**Fig. 5 The O-GlcNAcylation of LARS1 regulates ULK1-mediated LARS1 phosphorylation. a** SW620 cells were starved and stimulated with 11 mM glucose for the indicated durations. **b** Quantification of LARS1 S720 phosphorylation and O-GlcNAcylation of **a** (mean ± SEM, n = 3 independent experiments). **c**, **d** SW620 cells were transfected with the indicated siRNAs. After 48 h, the cells were starved of glucose for 4 h, pre-incubated with streptolysin O (SLO) for 5 min, and supplemented with 11 mM glucose or 200 μM indicated metabolites for 30 min. **e** SW620 cells were transfected with the indicated siRNAs. After 48 h, the cells were starved of glucose for 4 h with vehicle or indicated compounds (10 μM Compound C, 20 μM SBI-0206965, 1 μM LYN1604, 1 mM AICAR) **f** SW620 cells were transfected with the indicated LARS1 constructs. After 24 h, the cells were starved of glucose for 4 h and supplemented with 11 mM glucose for 30 min. **g** SW620 cells were transfected with control or siRNA targeting OGT1. After 48 h, the cells were starved of glucose at the indicated durations. **h** SW620 cells were transfected with siRNA targeting ULK1. After 24 h, the cells were transfected with the indicated expression construct. After 24 h, the cells were starved of glucose for 4 h. Each sample was analyzed by immunoblotting with the indicated antibodies. **i** SW620 control or S1042A knock-in cells were starved of glucose for 4 h and supplemented with glucose for 30 min. **a**, **c**, **d**, **e**, **f**, **g**, **i** Each sample was subjected to immunoprecipitation with sWGA-conjugated agarose beads, anti-LARS1 antibody-conjugated beads or anti-myc antibody-conjugated agarose beads and analyzed by immunoblotting with the indicated antibodies. Representative data of three experiments with similar results. Source data are provided as a Source Data file.

**The O-GlcNAcylation of LARS1 controls leucine-derived ATP production and protects cells under glucose starvation.** We demonstrated above that LARS1 S720 phosphorylation is induced by O-GlcNAcylation of LARS1 S1042 (Fig. 5f, g) and we know that this phosphorylation prevents leucine binding to the active site of LARS1 in the absence of glucose[34], when leucine degradation fuels oxidative phosphorylation (OXPHOS) in cells[34,47]. First, we confirmed that both O-GlcNAcylation and phosphorylation of LARS1 occurred under glucose starvation in various cell lines. Glucose starvation significantly increased the O-GlcNAcylation and S720 phosphorylation of LARS1 in all tested cell lines (Supplementary Fig. 6). While overexpression of the S1042A mutant increased S6K T389 phosphorylation even in the absence of glucose, overexpression of the S720D mutant suppressed mTORC1 activity even under glucose supplementation (Supplementary Fig. 6).

To investigate the metabolic consequences of leucine failing to bind LARS1 during glucose starvation, we expressed WT, S1042A mutant and S720D, S1042A double mutant LARS1 in these cells, starved them of glucose and leucine for 4 h then added back leucine for 4 h before assaying the oxygen consumption rate (OCR) of the cells. As expected, leucine increased the OCR in control cells and cells overexpressing WT LARS1 under glucose starvation (Fig. 6a). By contrast, in cells expressing the S1042A mutant, leucine supplementation did not increase OXPHOS under glucose starvation (Fig. 6a). Consistent with this, leucine supplementation did not increase OXPHOS also in LARS1 S1042A knock-in cells under glucose starvation (Supplementary Fig. 7a). In cells expressing the S720D, S1042A double mutant LARS1, the leucine-induced OXPHOS response was rescued even in the absence of glucose (Fig. 6a). These data suggest that LARS1-dependent leucine consumption for protein synthesis may be a crucial determinant of leucine-mediated mitochondrial OXPHOS.

Intracellular leucine is transported into mitochondria and converted to α-ketoisocaproic acid (KIC) by branched-chain amino acid transaminase 2 (BCAT2). Then, branched-chain α-ketoacid dehydrogenase complex (BCKD) catalyzes KIC into isovaleryl-CoA generating NADH, and in turn, isovaleryl-CoA is converted to 3-methylcrotonyl-CoA producing $FADH_2$ (Fig. 6b)[48,49]. NADH and $FADH_2$ are key molecules for OXPHOS transferring electron to electron transport chain complexes thereby synthesizing ATP. To investigate the effect of LARS1 O-GlcNAcylation on the generation of NADH from leucine catabolic pathway under glucose starvation, we measured the $NADH/NAD^+$ ratio in cells expressing WT, S1042A single mutant or S720D, S1042A double mutant LARS1. Cells were starved of glucose and leucine for 4 h then leucine was added for 4 h. Consistent with the OCR results, leucine resupplementation increased $NADH/NAD^+$ ratio in control and WT- and S720D, S1042A mutant-transfected cells but not S1042A mutant-transfected cells under glucose starvation (Fig. 6c). However, BCAT2 depletion significantly suppressed leucine-induced increase of $NADH/NAD^+$ ratio (Fig. 6c). In addition, incubation with cell-permeable KIC restored $NADH/NAD^+$ ratio reduced by BCAT2 downregulation under glucose starvation (Fig. 6c). To determine the effect of LARS1 O-GlcNAcylation on the production of ATP from leucine in the absence of glucose, we measured the intracellular ATP level. Similar with the NADH results, leucine resupplementation increased ATP production in control and WT- and S720D, S1042A double mutant-transfected cells but not S1042A single mutant-transfected cells (Fig. 6d). However, BCAT2 downregulation significantly suppressed leucine-induced ATP production, supporting the importance of leucine in the production of ATP under glucose-limited conditions (Fig. 6d). In addition, KIC rescued ATP production suppression induced by BCAT2 downregulation (Fig. 6d), indicating that the conversion of leucine to KIC is critical for energy production under glucose-limited conditions. Correspondingly, the addition of leucine ameliorated the extent of cell death due to glucose starvation in cells expressing WT LARS1 and the S720D, S1042A double mutant but not in cells expressing the S1042A single mutant. (Fig. 6e). Depletion of BCAT2 reduced this protective effect of leucine against cell death due to glucose starvation and KIC rescued this effect of BCAT2 depletion even in S1042A LARS1-transfected cells in the absence of glucose and leucine (Fig. 6e).

To determine whether the LARS1 S1042A-induced dysregulation of leucine catabolism is the major cause of cell death under glucose starvation, we measured the cell death in cells expressing LARS1 WT or S1042A mutant. Cells were starved of glucose and leucine for 4 h then leucine was added for 4 h under KIC, rapamycin, or cycloheximide. KIC treatment dramatically reduced the cleavage of caspase-3 and PARP caused by LARS1 S1042A expression in the absence of glucose without any effect on S6K T389 phosphorylation (Fig. 6f), suggesting that the major cause of S1042A LARS1-induced cell death was an ATP production crisis resulting from impaired leucine catabolism. Rapamycin and cycloheximide suppressed global protein synthesis analyzed by puromycin incorporation assay and also differentially suppressed the cleavage of caspase-3 and PARP caused by LARS1 S1042A in the presence of leucine and in the absence of glucose (Fig. 6f), suggesting that the inhibition of the dysregulated protein synthesis induced by S1042A can rescue ATP production crisis and cell death. Similarly, LARS1 S1042A knock-in cells displayed reduced response to leucine with decreased $NADH/NAD^+$ ratio, ATP levels, and increased cell death compared to control cells in the absence of glucose (Supplementary Fig. 7b–d). KIC treatment also significantly rescued these outcomes induced by LARS1 S1042A mutation (Fig. 6g and Supplementary Fig. 7b–d). These results suggest that dysregulated mTORC1 activity and protein synthesis, and

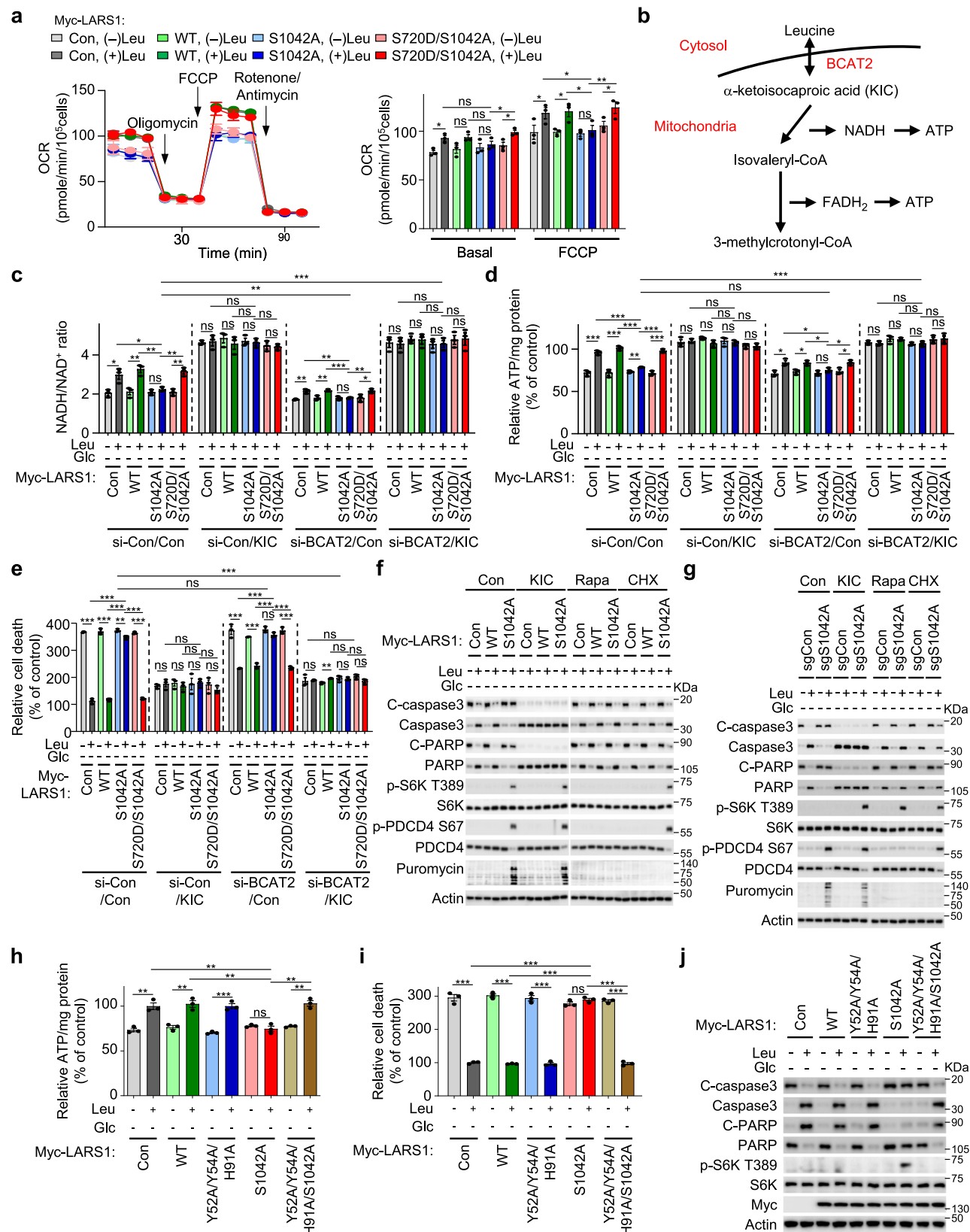

impaired leucine catabolism by the S1042A mutant collectively contributed to cell death under glucose starvation.

To investigate whether reduced leucine-binding by LARS1 might explain the increased leucine catabolism observed in the absence of glucose, we introduced a triple mutant of LARS1 (Y52A, Y54A, H91A), which has a reduced affinity for

leucine[14,43,50], and monitored leucine-induced S6K T389 phosphorylation, ATP production, and cell death in the absence of glucose. As predicted, this triple mutant lost the ability to activate mTORC1 in response to leucine and suppressed the ability of LARS1 S1042A to activate mTORC1 in the presence of glucose (Supplementary Fig. 7e) or the absence of glucose

**Fig. 6 The *O*-GlcNAcylation of LARS1 controls leucine-derived ATP production and protects cells under glucose starvation. a, c, d, e** SW620 cells were transfected with the indicated LARS1 constructs. After 24 h, the cells were starved of glucose and leucine for 4 h then leucine was added for 4 h. **a** cells were exposed to 2 μM oligomycin, 0.5 μM FCCP, and 0.5 μM/0.5 μM rotenone/antimycin, and the OCR was measured. Left: OCR over time; right: bar graph of basal OCR and FCCP-treated maximal OCR from left (mean ± SEM, n = 3 independent experiments). **b** Summary of the leucine catabolism pathway. **c, d, e** cells were harvested and analyzed with **c** NADH/NAD⁺ assay kit and **d** ATP assay kit. **e** cells were incubated with CellTox^TM Green dye and dead cells were detected by a live-cell imaging analyzer. (mean ± SEM, n = 3 independent experiments). **f** SW620 cells were transfected with the indicated LARS1 constructs. After 24 h, the cells were starved of glucose and leucine for 4 h then 0.4 mM leucine was added for 4 h with vehicle or indicated compounds: KIC, 200 μM α-ketoisocaproic acid; Rapa, 10 nM rapamycin; CHX, 20 μM cycloheximide. **g** SW620 control or S1042A knock-in cells were starved of glucose and leucine for 4 h then 0.4 mM leucine was added for 4 h. **f, g** 1 μM puromycin was added to the medium of the cells for 30 min. **h, i, j** SW620 cells were transfected with the indicated LARS1 constructs. After 24 h, the cells were starved of glucose and leucine for 4 h then leucine was added for 4 h. Each sample was harvested and analyzed with **h** ATP assay kit and **i** CellTox^TM Green dye signals (mean ± SEM, n = 3 independent experiments). **j** During leucine re-supplementation, vehicle or 1 μM puromycin was also added and incubated for 30 min. *P*-value was determined by two-tailed unpaired Student's *t* test. *, *P* < 0.05; **, *P* < 0.01; ***, *P* < 0.001; ns, not significant. Representative data of three experiments with similar results. Source data are provided as a Source Data file.

(Supplementary Fig. 7f). Consistent with this, combination of the Y52A, Y54A and H91A mutations with the S1042A mutation in LARS1 restored ATP synthesis from leucine in the absence of glucose (Fig. 6h) and reduced cell death caused by impaired *O*-GlcNAcylation in cells expressing LARS1 S1042A (Fig. 6i, j). We conclude from these data that *O*-GlcNAcylation of LARS1 is crucial in deciding whether leucine is catabolized to support cell survival when glucose is limited.

## Discussion

Here, we suggest a mechanistic model by which the leucine sensor LARS1 also senses decreased glucose levels and mediates defense under glucose starvation by modulating leucine metabolism. Under glucose limitation, LARS1 is *O*-GlcNAcylated and subsequently phosphorylated. These posttranslational modifications of LARS1 consequently inhibit the leucine-sensing Rag GTPase-mTORC1 axis and protein synthesis in cells. However, a lack of *O*-GlcNAcylated LARS1 consistently activates mTORC1 signaling and protein synthesis. Finally, *O*-GlcNAcylation of LARS1 changes the metabolic fate of leucine toward the catabolic pathway, protecting cells from death (Supplementary Fig. 8).

*O*-GlcNAcylation functions to regulate processes in response to nutrients and cellular stresses. When cells meet almost any type of stress conditions, such as glucose starvation, hypoxia, and oxidative stress, the *O*-GlcNAcylation of a large amount of proteins quickly increases[28,31,40]. Although serine or threonine phosphorylation is performed by a variety of kinases with substrate selectivity[51], mammalian cells possess only a single gene encoding the OGT1 catalytic subunit, implying that the *O*-GlcNAcylation of target proteins by OGT1 is mediated in a way similar to that of phosphatases[52]. Strikingly, while the substrate specificity of OGT1 is sensitive to the concentration of UDP-GlcNAc[53] and glucose starvation indeed decreases UDP-GlcNAc concentrations[29], glucose starvation paradoxically increases global protein *O*-GlcNAcylation, which is in agreement with other reports[29–31,40]. *O*-GlcNAcylation and phosphorylation extensively interact, with *O*-GlcNAcylation serving as a stress sensor to regulate cellular functions[54]. While protein phosphorylation exquisitely controls specific enzyme activity, *O*-GlcNAcylation broadly tunes intracellular processes in response to nutrient levels and cellular stress[55]. Similarly, *O*-GlcNAcylation of LARS1 was modified first, after which phosphorylation occurred, leading to the finely tuned adjustment of leucine metabolism. The *O*-GlcNAcylation and phosphorylation of LARS1 changed the fate of intracellular leucine from an anabolic process to a catabolic process, activating the leucine degradation pathways. These findings are an excellent example of the intimate crosstalk between these two most abundant modifications in the control of cellular metabolism.

Since glucose is an essential nutrient for cell survival, intracellular sensors must monitor glucose levels, affording adaptive responses to changes in glucose availability. The hub effector for these responses is the mTORC1 complex[56], and several studies have reported different glucose sensors that regulate mTORC1 activity in response to the environmental glucose level. AMPK is a well-known regulator of mTORC1 activity in response to glucose availability[21]. AMPK senses the concentrations of AMP under glucose starvation, and the displacement of ATP by AMP dramatically increases its activity[57]. Mechanistically, during glucose starvation, AMP binding to AMPK enhances the formation of the AXIN-LKB1-AMPK complex[58] and anchors the complex on the lysosomal membrane via Ragulator with inactivation of mTORC1[59]. Recently, the glycolytic enzyme aldolase was reported as a sensor for glucose availability that controls AMPK[60]. Under limited glucose conditions, FBP-unoccupied aldolase suppresses the v-ATP-Ragulator complex and promotes the formation of a complex including AXIN-LKB1-AMPK, triggering the T172 phosphorylation of AMPK[24]. AMPK stimulation switches on several catabolic, nutrient-scavenging processes, including autophagy, and ULK1 is a well-known target of AMPK[35,36,61]. In this context, ULK1 phosphorylates several target proteins, including LARS1, at residues crucial for leucine binding to support cell survival under glucose starvation[34]. Therefore, LARS1 receives at least two modifications for decreased cellular glucose levels under limited glucose conditions: one is OGT1-mediated S1042 *O*-GlcNAcylation, and the other is S720 phosphorylation by the aldolase-AMPK-ULK1 axis. Indeed, we observed that aldolase and its upstream and downstream substrates, FBP and DHAP, have the regulatory role of the LARS1-mediated mTORC1 activity. We also found that LARS1 S1042 *O*-GlcNAcylation was exclusively controlled by OGT1. These results imply that cells can finely control their metabolism and signaling pathways through multiple glucose sensors to respond to nutrient stress conditions such as glucose starvation.

In conclusion, our data suggest that LARS1 coordinates leucine- and glucose-sensing pathways and determines the direction of leucine metabolism based on glucose availability.

## Methods

**Cell culture**. SW620 (#CCL-227), SW480 (#CCL-228), A549 (#CRL-185), PANC1 (#CRL-1469), HeLa(#CCL-2), and RD(#CCL-136) were obtained from the American Type Culture Collection (ATCC). HEK293T (#21573) was obtained from the Korean Cell Line Bank. AMPKγ1 WT or AMPKγ1 KO 293 A cell line was a kind gift of Prof. Hyun Woo Park (Yonsei University, Seoul, Korea). SW620, SW480, and A549 cells were cultured in RPMI-1640 medium (Welgene, #LM011-01) supplemented with 10% heat-inactivated fetal bovine serum (FBS) (Welgene, #S001-07) and 1% penicillin-streptomycin (Hyclone, #SV30010) in a 37 °C incubator in a 5% CO₂ atmosphere. HEK293T, PANC1, RD, AMPKγ1 WT 293 A, AMPKγ1 KO 293 A, and HeLa cells were cultured in DMEM medium (Welgene,

#LM001-05) supplemented with 10% heat-inactivated FBS and 1% penicillin-streptomycin in a 37 °C incubator in a 5% $CO_2$ atmosphere.

**Antibodies and compounds**. The following antibodies were obtained from the following sources: antibodies against phospho-p70 S6 kinase (T389)(WB dilution 1:1000, Cell signaling, #9205), p70 S6 kinase (WB dilution 1:1000, Cell signaling, #9202), phospho-4EBP1 (Thr37/46) (236B4) (WB dilution 1:1000, Cell signaling, #2855), 4EBP1 (53H11) (WB dilution 1:1000, Cell signaling, #9644), RagC (D8H5) (WB dilution 1:1000, Cell signaling, #9480), RagB (D18F3) (WB dilution 1:1000, Cell signaling, #8150), mTOR (7C10) (WB dilution 1:1000, Cell signaling, #2983), Raptor (24C12) (WB dilution 1:1000, Cell signaling, #2280), Hexokinase II (C64G5) (WB dilution 1:1000, Cell signaling, #2867), Aldolase A (D73H4) (WB dilution 1:1000, Cell signaling, #8060), DYKDDDDK Tag (Binds to same epitope as Sigma's Anti-FLAG® M2 Antibody) (9A3) (WB dilution 1:1000, Cell signaling, #8146), phospho-AMPKα (T172), (40H9) (WB dilution 1:1000, Cell signaling, #2535), AMPKα (WB dilution 1:1000, Cell signaling, #2532), AMPKγ (WB dilution 1:1000, Cell signaling, #4187), phospho-ULK1 (S757) (WB dilution 1:1000, Cell signaling, #6888), phospho-Atg13 (S355) (D6J1W) (WB dilution 1:1000, Cell signaling, #26839), Atg13 (D4P1K) (WB dilution 1:1000, Cell signaling, #13273), phospho-Atg14 (S29) (D4B8M) (WB dilution 1:1000, Cell signaling, #92340), Atg14 (D1A1N) (WB dilution 1:1000, Cell signaling, #96752), LARS1 (WB dilution 1:1000, Cell signaling, #13868), RagD (WB dilution 1:1000, Bethyl Laboratories, #A304-301A), LARS1 (WB dilution 1:1000, IF dilution 1:200, IP: 2ug/400ug protein, Bethyl Laboratories, #A304-315A), LAMP2 (H4B4) (WB dilution 1:1000, IF dilution 1:50, Santa cruz, #c-18622), ARF1 (ARFS 1A9/5) (WB dilution 1:1000, Santa cruz, #sc-53168), b-actin (C4) (WB dilution 1:1000, Santa cruz, #sc-47778), c-Myc (9E10) (WB dilution 1:1000, IP: 2ug/300ug protein, Santa cruz #sc-40), O-GlcNAc (RL2), (WB dilution 1:1000, Santa cruz, #sc-59624), HA-Tag (F-7) (WB dilution 1:1000, Santa cruz, #sc-7392), ULK1 (H-240) (WB dilution 1:1000, Santa cruz, #sc-33182), OGT1 (WB dilution 1:1000, IP: 2ug/400ug protein, abcam #ab96718), phospho-PDCD4 (S67) (WB dilution 1:1000, abcam, #ab73343), PDCD4 (WB dilution 1:1000, abcam, #ab80590), Aldolase B (WB dilution 1:8000, Proteintech, #18065-1-AP), Aldolase C (WB dilution 1:5000, Proteintech, #A-11001), Sestrin 2 (WB dilution 1:2000, Proteintech, #10795-1-AP), EPRS1 (WB dilution 1:5,000, Neomics, #NMS-01-0004), Alexa488-conjugated secondary antibody (IF dilution 1:500 Invitrogen #14884-1-AP), Alexa594-conjugated secondary antibody (IF dilution 1:500 Invitrogen #A-11012), Anti-mouse IgG, HRP secondary antibody (WB dilution 1:10000 Invitrogen #31430), and anti-rabbit IgG, HRP secondary antibody (WB dilution 1:10000 Invitrogen #31460). The following reagents were obtained from the following sources: Rapamycin (Sigma-Aldrich, #171260), 2-Deoxy-D-glucose (Sigma-Aldrich, #D8375), DAPI (Sigma-Aldrich, #D9542), Dihydroxyacetone phosphate dilithium salt (DHAP) (Sigma-Aldrich, #D7137), Compound C (Sigma-Aldrich, #171260), AICAR (Sigma-Aldrich, #A9978), SBI-0206965 (Sigma-Aldrich, #SML1540), Imidazole (Sigma-Aldrich, #I2399), Streptolysin O (Sigma-Aldrich, #S5265), UDP-GlcNAc (N-Acetyl-D-glucosamine) (Sigma-Aldrich, #A8625), Thiamet G (Sigma-Aldrich, #SML0244), LYN-1604 (Cayman #24007), D-fructose-1,6-bisphosphate (FBP) (Cayman #20516), ST045849 (R&D Systems, #6775), and 4-methyl-2-oxopentanoic acid (KIC) (MedChemExpress, #HY-W012722).

**Transfection with cDNA constructs and siRNAs**. Cells were grown at $1 \times 10^6$ cells/well in 6-well plates. Twenty-four hours after incubation, the cells were transfected with cDNA constructs by using TurboFect transfection reagent (Thermo Fisher Scientific, # R0531). Twenty-four hours after transfection, the cells were used in the experiments performed in this study. For siRNA transfection, cells were grown at $2 \times 10^5$ cells/well in 6-well plates. Twenty-four hours after incubation, the cells were transfected with siRNA by using Lipofectamine 2000 reagent (Thermo Fisher Scientific, #11668500). Forty-eight hours after transfection, the cells were used for the experiments. The following siRNAs were used and purchased from Invitrogen, Thermo Scientific, Integrated DNA technologies. Control siRNA (Invitrogen, #AM4611), Human BCAT2 siRNA (Invitrogen, #BCAT2HSS1841846431950), Human Sestrin1 siRNA (Thermo Scientific, #HSS120529), Human Sestrin2 siRNA (Thermo Scientific, # HSS130295), Human ALDOA siRNA (Integrated DNA technologies, #hs.Ri.ALDOA.13.1), Human ALDOB siRNA (Integrated DNA technologies, #hs.Ri.ALDOB.13.1), Human ALDOC siRNA (Integrated DNA technologies, #hs.Ri.ALDOC.13.1), Human Hexokinase II siRNA (Integrated DNA technologies, # hs.Ri.HK2.13.1), Human G6PD siRNA (Integrated DNA technologies, # hs.Ri.G6PD.13), Human PHGDH siRNA (Integrated DNA technologies, # hs.Ri.PHGDH.13). The following siRNAs were used and purchased from Integrated DNA technologies (IDT) (Supplementary Table 1).

**Nutrient starvation and stimulation**. For glucose starvation, cells were rinsed twice with PBS and incubated in glucose-free DMEM or RPMI-1640 medium (Welgene, # LM001-79, #LM011-60) with 10% dialyzed FBS (Gibco, #26400044) for 4 h. The cells were restimulated with DMEM or RPMI-1640 medium containing glucose for 30 min. For leucine starvation, cells were rinsed twice with PBS and incubated in leucine-free DMEM or RPMI-1640 medium (Welgene, #LM001-91, #LM011-96) with 10% dialyzed FBS for 1.5 h. The cells were restimulated with

DMEM or RPMI-1640 medium containing the indicated concentrations of leucine for 15 min. For the leucine and glucose stimulation assays, the cells were incubated in glucose-free DMEM or RPMI-1640 medium for 2.5 h, followed by incubation in glucose- and leucine-deficient DMEM or RPMI-1640 medium (Welgene, #LM001-91, #LM011-82) for 1.5 h. Four hours after starvation, the cells were restimulated with leucine for 15 min or glucose for 30 min based on the experimental design.

**Cell growth, viability, and death assays**. SW620 cells were seeded in 96-well plates and incubated for 24 h. After compound treatment or nutrient starvation, the cell culture medium was exchanged with medium containing CellTox™ Green Dye (Promega, #G8741). Phase and green fluorescence images were acquired by using IncuCyte™ Zoom (Essen BioScience). Quantitative analysis was performed by using the IncuCyte™ Zoom basic analyzer. The presence of a green color (CellTox™ object count/mm² over time) was used to quantify the number of dead cells, and phase (CellTox™ object count/mm² over time) was used to quantify the cell viability.

**Immunoblotting and immunoprecipitation**. Cells were rinsed with PBS and harvested using lysis buffer containing 50 mM Tris-HCl (pH 7.4), 150 mM NaCl, 5 mM EDTA, 0.5 mM EGTA, 10% glycerol, 0.5% Triton X-100, and 1 mM phenylmethylsulfonyl fluoride. Each cell lysate was centrifuged at 17,700 × g and 4 °C for 20 min, and the supernatant was used for immunoblotting. For immunoprecipitation, cells were lysed in buffer containing 50 mM Tris-HCl (pH 7.4), 10 mM NaCl, 1 mM EDTA, 0.5 mM EGTA, 1 mM $MgCl_2$, 0.5% Triton X-100, and 1 mM phenylmethylsulfonyl fluoride. Each cell lysate was centrifuged at 17,700 × g and 4 °C for 20 min, and the supernatant was used for reaction with antibodies. Primary antibodies were added to the lysates and incubated with rotation overnight at 4 °C. Protein A/G agarose beads (Santa Cruz, #sc-2003) were added and incubated for 4 h at 4 °C. After washing five times with lysis buffer containing phosphatase inhibitor and protease inhibitor, the precipitates were dissolved in 2.5× sample buffer containing 30 mM Tris-HCl (pH 6.8), 12.5% glycerol, 1% SDS, 7.2 mM 2-mercaptoethanol, and 0.1% bromophenol blue and separated by SDS-PAGE. Then, the proteins were transferred to PVDF membranes (Merck, #IPVH00010) using a Trans-Blot Turbo Blotting System (Bio-Rad). After blocking in TBST buffer containing 5% BSA or 5% skim milk, the membranes were incubated with individual primary antibodies overnight. The next day, the membranes were subsequently incubated with either anti-mouse or anti-rabbit IgG conjugated with horseradish peroxidase. Immunoblot signals were detected by MicroChemi (DNR Bioimaging system) with enhanced chemiluminescence, EzWestLumi (ATTO, #AE-1495) and quantified by densitometry analysis of protein bands using Multi Gauge 3.0.

**Precipitation with sWGA-conjugated beads**. Cells were rinsed twice with PBS and lysed in lysis buffer containing 50 mM Tris-HCl (pH 7.4), 150 mM NaCl, 5 mM EDTA, 0.5 mM EGTA, 10% glycerol, 0.5% Triton X-100, and 1 mM phenylmethylsulfonyl fluoride. After the lysates were centrifuged at 17,700 × g and 4 °C for 20 min, each sample was incubated with sWGA-conjugated beads (Vector Laboratories, # AL-1023S) for 12 h. After washing five times with lysis buffer, sWGA-bound proteins were eluted with lysis buffer containing 2.5× sample buffer and used for immunoblotting.

**GTP-agarose bead pulldown assay**. Cells were rinsed twice with PBS, lysed with GTP-binding buffer (20 mM Tris-HCl pH 7.5, 5 mM $MgCl_2$, 2 mM PMSF, 20 µg/mL leupeptin, 10 µg/mL aprotinin, 150 mM NaCl and 0.1% Triton X-100) and sonicated for 15 sec. After the lysates were centrifuged at 17,700 × g and 4 °C for 20 min, the supernatants were collected. Each sample was incubated with 100 µl of GTP-agarose beads (Sigma-Aldrich, #G9768) overnight at 4 °C. After washing five times with GTP-binding buffer, GTP-bound protein extracts were eluted with 2.5× sample buffer.

**His-LARS1 purification with Ni-NTA agarose beads**. pQE80l-WT LARS1, pQE80l-S1042A LARS1 mutant, pETDeuT1 WT LARS1 with OGT1, and pET-DeuT1 S1042A LARS1 mutant with OGT1 were transformed into E. coli (BL21(DE3) chemically competent E. coli, enzynomics, #CP111) and a single colony of these cells was grown in 3 ml of LB medium containing 100 µg/ml ampicillin at 37 °C overnight. Then, each sample was transferred to 400 ml of LB medium with ampicillin for growth until the cell density reached an OD600 between 0.5~0.7. Thereafter, 0.5 mM IPTG (Sigma-Aldrich, #I6758) was added to induce protein expression at 16 °C overnight. The cells were harvested by centrifugation at 17,700 × g and 4 °C for 30 min, and the cell pellets were resuspended in 5 ml of lysis buffer containing 50 mM Tris-HCl (pH 7.4), 0.5 M NaCl, 2 mM EDTA, 2 mM β-mercaptoethanol, 20 mM imidazole (Sigma-Aldrich, #I2399), and protease inhibitor cocktail (Sigma-Aldrich, #P2714) on ice for 10 min, after which the cell lysate was sonicated for 30 sec 5 times. The cell lysates were centrifuged at 17,700 × g and 4 °C for 20 min, and the supernatant was transferred to a new 15-ml tube. Samples were applied to a disposable 5-ml-polypropylene column with Ni-NTA beads (Thermo Fisher Scientific, #R90101). After washing five times with lysis buffer containing 50 mM imidazole, His-bound proteins were eluted with lysis buffer containing 250 mM imidazole. Finally, the proteins were dialyzed overnight

against buffer containing 50 mM HEPES/NaOH (pH 7.4), 150 mM NaCl, 5 mM MgCl2, 0.5 mM EDTA, and 0.1% Triton X-100 at 4 °C.

**In vitro pulldown assay**. His-tagged WT LARS1 and S1042A LARS1 were purified and incubated with RagD$^{GTP}$- or ULK1-expressing cell lysate in the absence or presence of 200 µM ATP and 2 mM L-leucine for 1 h and then pulled down with Ni-NTA agarose beads. The binding assay was conducted in buffer containing 50 mM HEPES/NaOH (pH 7.4), 150 mM NaCl, 5 mM MgCl2, 0.5 mM EDTA, and 0.1% Triton X-100.

**In vitro GTPase assay**. His-tagged WT LARS1 and S1042A LARS1 were purified, and then incubated with RagD$^{WT}$- or RagC$^{WT}$-expressing cell lysate in the absence or presence of 200 µM ATP and 2 mM L-leucine for 1 h and then pulled down with GTP-agarose beads. After washing 5 times with GTP-binding buffer, the precipitated proteins were eluted with a 2.5× sample buffer.

**GST-LARS1 purification and In vitro pulldown assay**. pGEX4T-1-WT LARS1 (aa 1-1176), pGEX-4T1-S1042A LARS1 (aa1-1176), pGEX4T1-WT LARS1 (aa 1-1062), pGEX4T1-S1042A LARS1 (aa 1-1062), and pGEX4T1-WT LARS1 (aa 1-892) were transformed into E. coli (BL21), and a single colony of these cells was grown in 3 ml of LB medium containing 100 µg/ml ampicillin at 37 °C overnight. Then, each sample was transferred to 400 ml of LB medium with ampicillin for growth until the cell density reached an OD600 between 0.5~0.7. Thereafter, 0.5 mM IPTG (Sigma-Aldrich) was added to induce protein expression at 16 °C overnight. The cells were harvested by centrifugation at 17,700 × g and 4 °C for 30 min, and the cell pellets were resuspended in 5 ml of lysis buffer containing 50 mM HEPES/NaOH (pH 7.4), 150 mM NaCl, 5 mM MgCl2, 0.5 mM EDTA, and 0.1% Triton X-100, and protease inhibitor cocktail (Sigma-Aldrich) on ice for 10 min, after which the cell lysate was sonicated for 30 sec 5 times. The cell lysates were centrifuged at 17,700 × g and 4 °C for 20 min, and the supernatant was transferred to a new 15-ml tube. Samples were applied to a disposable 5-ml-polypropylene column with glutathione Sepharose 4B (Thermo Fisher Scientific, #16100). After washing five times with buffer (50 mM HEPES/NaOH (pH 7.4), 150 mM NaCl, 5 mM MgCl$_2$, 0.5 mM EDTA, and 0.1% Triton X-100), GST-tagged WT LARS1 and S1042A LARS1 proteins were incubated with ULK1 expressing cell lysate. The binding assay was conducted in buffer containing 50 mM HEPES/NaOH (pH 7.4), 150 mM NaCl, 5 mM MgCl$_2$, 0.5 mM EDTA, and 0.1% Triton X-100.

**Strep-LARS1 purification with strep-agarose beads**. Cells were rinsed twice with PBS and lysed in lysis buffer containing 50 mM Tris-HCl (pH 7.4), 150 mM NaCl, 5 mM EDTA, 0.5 mM EGTA, 10% glycerol, 0.5% Triton X-100, and 1 mM phenylmethylsulfonyl fluoride. After the lysates were centrifuged at 17,700 × g and 4 °C for 20 min, each sample was incubated with strep beads (Strep-Tactin Sepharose Suspension, IBA Lifesciences, #2-1201-002) for 16 h. After that, the samples were applied to a disposable 5-ml polypropylene column (Thermo Fisher Scientific, #29922). After washing five times with Buffer W (Strep Tag Washing Buffer, IBA Life Sciences, #2-1003-100), strep-bound proteins were eluted with buffer BXT (D-Desthiobiotin Buffer E, IBA Life Sciences, #2-1000-025).

**Mass spectrometry analysis**. Purified strep-LARS1 WT or S1042A sample was separated using SDS-PAGE and Coomassie stained with Imperial™ Protein Stain (Thermo), respectively. For LC-MS/MS analyses, the gel was de-stained and bands cut and processed as follows. Briefly, purified proteins bands were divided into 10 mm sections and subjected to in-gel digestion with trypsin. The tryptic digests were separated by online reversed-phase chromatography using a Thermo Scientific Eazy nano LC 1200 UHPLC equipped with an autosampler using a reversed-phase peptide trap Acclaim PepMap™ 100 (75 µm inner diameter, 2 cm length) and a reversed-phase analytical column PepMap™ RSLC C18 (75 µm inner diameter, 15 cm length, 3 µm particle size), both from Thermo Scientific, followed by electrospray ionization at a flow rate of 300 nl min−1. Samples were eluted using a split gradient of 3-50% solution B (80% ACN with 0.1% FA) in 60 min and 50–80% solution B in 10 min followed column wash at 100% solution B for 10 min. The chromatography system was coupled in line with an Orbitrap Fusion Lumos mass spectrometer. The mass spectrometer was operated in a data-dependent mode with the 120,000 resolution MS1 scan (375–1500 m/z), AGC target of 5e5 and max injection time of 50 ms. Peptides above threshold 5e3 and charges 2–7 were selected for fragmentation with dynamic exclusion after 1 time for 15 s and 10 ppm tolerance. Tandem MS (MS / MS) was performed with the quadrupole for HCD (Collision energy 28%, stepped collision energy 5%) or EThcD (maximum injection time 250 ms, 300,000 AGC target) and analyzed with a resolution of 30,000. All MS/MS spectra were analyzed using Sequest (XCorr Only) (Thermo Fisher Scientific, San Jose, CA, USA; version IseNode in Proteome Discoverer 2.2.0.388) and X! Tandem (The GPM, thegpm.org; version CYCLONE (2010.12.01.1)). Sequest (XCorr Only) was set up to search LARS_human.fasta (unknown version, 1 entries) assuming the digestion enzyme trypsin. X! Tandem was set up to search a reverse concatenated subset of the LARS_human database (unknown version, 2 entries) also assuming trypsin. Carbamidomethyl of cysteine was specified in Sequest (XCorr Only) and X! Tandem as a fixed modification. Glu->pyro-Glu of

the n-terminus, ammonia-loss of the n-terminus, gln->pyro-Glu of the n-terminus, oxidation of methionine, acetyl of lysine, phospho of serine, dicarbamidomethyl of lysine and hexNAc of serine were specified in X! Tandem as variable modifications. Oxidation of methionine, acetyl of lysine, phospho of serine, GG of lysine and hexNAc of serine were specified in Sequest (XCorr Only) as variable modifications. For the PTM location validation, Sequest (XCorr Only) and X! Tandem were searched with a fragment ion mass tolerance of 0.100 Da and a parent ion tolerance of 10.0 PPM and finally selected with 95% peptide threshold. CRITERIA FOR PROTEIN IDENTIFICATION– Scaffold (version Scaffold_4.11.0, Proteome Software Inc., Portland, OR) was used to validate MS/MS based peptide and protein identifications. Peptide identifications were accepted if they could be established at greater than 95.0% probability. Peptide Probabilities from X! Tandem (sample Mudpit_Treated_LRS2: Treated_LRS1) were assigned by the Scaffold Local FDR algorithm. Peptide Probabilities from Sequest (XCorr Only) and X! Tandem (sample Mudpit_untreated_LRS2: untreated_LRS1) were assigned by the Peptide Prophet algorithm[62] with Scaffold delta-mass correction. Protein identifications were accepted if they could be established at greater than 95.0% probability and contained at least 2 identified peptides. Protein probabilities were assigned by the Protein Prophet algorithm[63]. Proteins that contained similar peptides and could not be differentiated based on MS/MS analysis alone were grouped to satisfy the principles of parsimony.

**Enzymatic labeling of O-GlcNAcylation site**. Enzymatic labeling of O-GlcNAcylated LARS1 was conducted via Click-iT™ O-GlcNAc Enzymatic Labeling System (Invitrogen, #C33368). Purified LARS1 WT or S1042A samples were incubated with UDP-GalNAz. Then, samples were incubated with Gal-T1 (Y289L) enzyme for 24 h at 4 °C. After incubation, samples were rinsed twice with reaction buffer to remove UDP-GalNAz. Samples were incubated with TAMRA (streptavidin) and detected with streptavidin-HRP antibody.

**Immunofluorescence staining**. HeLa cells were seeded onto coverslips and fixed with 100% methanol for 5 min at 4 °C. After washing five times with PBS, the cells were incubated with the primary antibody (LAMP2 1:100, LARS1 1:250) for 2 h. After primary antibody incubation, the cells were rinsed with PBS and incubated with Alexa 488- or Alexa 595-conjugated secondary antibody (1:500) for 30 min. Nuclei were stained with a DAPI solution. After washing three times with PBS, the cells were mounted and observed via Zen imaging software (Zeiss). Quantitative analysis of lysosomal colocalization was performed using regions of interest (ROIs) in Zen imaging software (Zeiss). After the ROIs were defined according to the localization of LARS1 and LAMP2, the localization of other components was measured with the defined ROIs. The index of LARS1 colocalization corresponds to the mean ± standard deviation (SD) of the overlap coefficient (R)*100 obtained for more than 10 cells for each colabeling experiment. An intensity profile was generated using the image profile module in Zen imaging software (Zeiss). All experimenters were blinded to the sample groups to avoid experimenter bias in the results.

**Lysosomal fractionation**. The lysosome fraction was obtained using a lysosome enrichment kit (Thermo Fisher Scientific, #89839) following the manufacturer's instructions with minor modifications. Cells were rinsed with PBS and lysed with Dounce homogenizers on ice. After the lysates were centrifuged at 500 × g and 4 °C for 20 min, the supernatant was added to a discontinuous density gradient consisting of 17, 20, 23, 27, and 30% OptiPrep. After ultracentrifugation of the supernatant at 145,000 × g for 2 h at 4 °C, the lysosome band at the top of the gradient was collected and analyzed.

**Mutagenesis of LARS1**. LARS1 mutants were generated via site-directed mutagenesis (INTRON, #15071) by using point mutation primers, and the mutations were confirmed by DNA sequencing. The following all primers for mutagenesis were used and purchased from cosmogenetech (Supplementary Table 2).

**Leucylation assay**. The leucylation assay was carried out in a reaction buffer (50 mM HEPES-KOH (pH 7.6), 25 mM KCl, 5 mM MgCl$_2$, 1 mM spermine, 4 mM ATP, 2 mg/ml yeast total tRNA, 1 µM [$^3$H] leucine (60 Ci/mmol)). Enzyme reaction was initiated by the addition of 100 nM purified His-LARS1 wild type or S1042A. After 5 min, enzyme reaction mixtures were quenched on Whatman filter paper presoaked with 5% trichloroacetic acid (TCA). The papers washed three times with 5% TCA for 10 min at 4 °C and incubated with 100% ethanol for 10 min. The papers were then dried and radioactivity was quantified in liquid scintillation counter.

**NADH/NAD$^+$ assay**. NADH/NAD$^+$ assay was carried out NADH/NAD$^+$ Assay kit (Colorimetirc, Abcam, #ab65348). Cells in 6-well were starved of glucose and leucine for 4 h then leucine was added for 4 h and harvested each sample. After the lysates were centrifuged at 400 × g and at 4 °C for 5 min, extract cells were incubated with 400 µl of NADH/NAD extraction buffer at 4 °C for 20 min and RT for 10 min. After samples were centrifuged at 17,700 × g and 4 °C for 5 min and collected supernatant and transferred into a new 1.5 ml tube at 4 °C for 10 min.

Samples were divided into two groups. One group was heated at 60 °C for 30 min and the other group was kept on ice. A total 20 µl of two groups were transferred to 96-well microplate. After addition of 20 µl of the enzyme reaction mixture, samples were measured at 460 nm by a plate reader (Infinite 200 Pro, Tecan).

**Generation of S1042A knock-in cell.** Single Guide RNA (sgRNA) and donor DNA were designed using invitrogen™ TrueDesign™ Genome Editor. sgRNA and donor DNA was selected by the given score of 95.07 and specified in invitrogen™ TrueDesign™ Genome Editor. 5' sgRNA sequence is 5'-GATTT-TATCTTCTGCTTCGG-3' and 5' donor DNA sequence is 5'-CTGTTTAGCTA-GAACACATAGAAGTCAAGTTTGC CCCTGAAGCAGAAGA-TAAAATCAGGGAAGACTGCTGTCCT-3'. SW620 cells were seeded in 24-well plates and incubated for 24 h. sgRNA was annealed with tracrRNA at 95 °C for 5 min and mixed with 25 µl serum-free RPMI1640 media, 1250 ng TrueCutTM Cas9 Protein v2 (Thermo Fisher Scientific, #A36496), 500 ng Donor DNA, 2.5 µl LipofectamineTM Cas9 PLUSTM Reagent (Thermo Fisher Scientific, #CMAX00001(100035635)). At the same time, 1.5 µl LipofectamineTM CRISPR-MAXTM Reagent (Thermo Fisher Scientific, #CMAX00001(100035630)) was diluted by 25 µl serum-free RPMI1640 medium. These mixtures were incubated for 1 min and mixed for 12 min at room temperature. The cells were transfected with a transfection complex. Forty-eight hours after transfection, cells were rinsed twice with PBS and incubated in RPMI-1640 medium with 1 µg/ml puromycin (Sigma-Aldrich) for 72 h. Transfected cells isolate single-cell colonies for genotyping in a 96 well plate. After 5 days change the medium to RPMI1640 with puromycin and split the colonies for cell culture and genotyping. For genotyping the generated S1042A knock-in cells, each genomic DNA from clones was isolated using a gDNA extraction kit (TaKaRa, #9765 A) according to the manufacturer's recommendations. To detect the S1042A mutation of LARS1, the first PCR round was undertaken in a final volume of 25 µl. PCR product was electrophoresed on a 2% TBE agarose and visualized with GelDoc system and was isolated using a DNA extraction kit (LaboPass, #CMG0112) according to the manufacturer's recommendation. To confirm the S1042A mutation in LARS1, Sanger sequencing was performed and analyzed.

**Measurement of the OCR.** The OCR was measured using an XFe24 extracellular flux analyzer (Agilent Technologies) as described in the manufacturer's protocol. A total of $2.5 \times 10^4$ cells were seeded per well in 24-well microcell culture plates (Agilent Technologies) in glucose- and leucine-free DMEM with 10% dialyzed FBS and incubated at 37 °C for 4 h in a 5% $CO_2$ incubator. After incubation, the DMEM with or without leucine was replaced with phenol red and bicarbonate-free medium, and the cells were incubated at 37 °C for 4 h in a non-$CO_2$ incubator to equilibrate the $CO_2$ level in the atmosphere. Using the XFe24 analyzer, the OCR was measured under baseline conditions and under treatment with several metabolic drugs: oligomycin (2 µM), carbonyl cyanide-p-trifluoromethoxyphenylhydrazone (FCCP, 0.5 µM), and rotenone/antimycin A (0.5 µM/0.5 µM) (these drugs were included in the Cell Mito Stress Test Kit, Agilent Technologies). Each measurement cycle consisted of 3 min of mixing, 3 min of waiting, and 4 min of measuring. The OCR value was normalized to the cell number and analyzed using WAVE software v2.6.0.31 (Agilent Technologies).

**ATP assay.** Cells in 24-well format were starved of glucose and leucine for 4 h then leucine was added for 4 h. After incubation, the cells were washed using 1 ml of PBS and lysed using 500 µl of lysis buffer. ATP was measured by mixing each 40-µl supernatant of the lysed sample and 40 µl of CellTiter-Glo (Promega, #G9241). The plate was incubated in the dark for 10 min, and luminescence was quantified by a plate reader (Infinite 200 Pro, Tecan).

**Circular dichroism analysis.** Near-UV circular dichroism spectra were obtained using a J-715 spectropolarimeter (JASCO) at the Korea Basic Science Institute. Circular dichroism measurements were performed at 25 °C using a quartz cell with a path length of 10 mm. The protein samples were prepared at 1 mg/ml in PBS. The spectra were recorded from 350 nm to 250 nm, and five scans were averaged. Data were recorded at a scan speed of 100 nm/min and bandwidth of 1.0 nm with a 1-s response and 0.1-nm resolution.

**Protein structure modeling.** The X-ray structure of LARS1 ("sensing-on" structure PDB entry: 6KQY, "sensing-off" structure PDB entry: 6KR7) was used to produce the O-GlcNAcylated LARS1 model. The O-GlcNAcylated serine residue was added according to the parameters for this type of residue from the Vienna-PTM server[64–66]. Each LARS1 structure was uploaded to the Vienna-PTM server and the 1042nd serine residue was selected to have O-GlcNAcyl residue added. This step made available force field parameters and frameworks via water molecule addition with pre-minimized coordinates. The energy minimization step was conducted for determination of proper local orientation. Vienna-PTM server presented with modified PDB file and protein coordinates. This protein coordinates obtained from this server was loaded into Maestro software (Schrödinger Release 2019-3), and the preparation wizard assigned bond orders with preprocess method, added hydrogen atoms within the dialog box which optimizes the hydrogen

bonding network on the hydroxyl and amide groups, and filled in missing side chains and loops. Default parameters were applied to optimize the hydrogen bond assignment via optimization operation. Molecular dynamics (MD) simulation was conducted via the Desmond default settings. Desmond application was opened and O-GlcNAcylated LARS1 protein was uploaded. Energy minimization was conducted by Desmond simulation. Structure relaxation was conducted by the relax panel of Desmond. Lastly, NVIDIA GeForce GTX 1660 Super was selected for the final MD simulation. The coordinates of the input and final protein structures are provided in the Source Data file. The initial configuration of the MD simulation was provided with the protein structure information, MD-1 PDB file, and the atom coordinate information, MD-1 TXT file. The final configuration of the MD simulation was provided with the protein structure information, MD-2 PDB file, and the atom coordinate information, MD-2 TXT file. The final protein structure was used for structural comparison (Schrödinger Release 2019-3). All structural analyses, such as structure superimposition and RMSD calculations between LARS1 structures, were performed with the Coot[67] and PyMOL (http://pymol.org/) programs. Structural flexibility was assessed by the residue fluctuation values, which were simulated using the CABS-flex server with the protein coordinates described above[68].

**Limited proteolysis.** To compare the digestion pattern of WT LARS1 and the S1042A LARS1, samples were prepared in reaction buffer containing 20 mM HEPES-KOH (pH 7.4), 150 mM KCl, and 10 mM $MgCl_2$. Proteinase K (PK, 1 mg/ml) was added to the purified proteins at a PK:protein ratio of 1:100 and incubated for 5 min at 25 °C. Proteolysis was stopped by the addition of sample buffer containing 5 mM PMSF and immediate heat inactivation for 6 min at 99 °C.

**Quantification and statistical analysis.** All statistical analyses were performed using GraphPad Prism 8 software. Statistical significance of data obtained from immunofluorescence staining, the cell viability assay, the NADH/NAD+ assay, and OCR measurements was determined by two-tailed unpaired Student's $t$-test. Data in bar graphs are shown as mean ± SEM or SD. Associated $P$ values are indicated as follows: *, $P < 0.05$; **, $P < 0.001$; ***, $P < 0.001$; ns, not significant $P > 0.05$. Exact $P$ values are provided in the Source Data file. No additional statistical tests for data distributions.

**Reporting Summary.** Further information on research design is available in the Nature Research Reporting Summary linked to this article.

## Data availability

The coordinates of the input and final protein structures of the MD simulations are provided in the Source Data file. The LARS1 crystal structures referenced in this study are available from the Protein Data Bank under the accession codes 6KQY, 6KR7. Source data are provided in the Source Data file. Other information needed is available from the corresponding author upon request. Source data are provided with this paper.

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

## Acknowledgements

This work was supported by the Basic Science Research Program through the National Research Foundation (NRF) of Korea funded by the Ministry of Education

(2018R1A6A1A03023718, 2020R1I1A1A01067423), the Bio & Medical Technology Development Program of the National Research Foundation (NRF) funded by the Ministry of Science & ICT (2020M3E5E2040282), and a National Research Foundation of Korea (NRF) grant funded by the Ministry of Science & ICT (MSIT) (2020R1A2C2099586, 2021R1C1C2006283). Y.S was supported by the Health Fellowship Foundation.

## Author contributions

K.K, and H.C.Y conceived and designed the experiment, collected and analyzed data, and wrote the manuscript; B.G.K and K.M performed mass spectrometry analysis. Sulhee Kim, and K.Y.H performed circular dichroism analysis and 3D structure modeling. Y.S, I.Y, Y.C.Y, S.J.P, J.H.K, and Sunghoon Kim, collected and analyzed the data; J.M.H, supervision, manuscript writing, and final approval of the manuscript.

## Competing interests

The authors declare no competing interests.
