## [Peer review file · Nature Communications]

REVIEWER COMMENTS

Reviewer #1 (Remarks to the Author):

Glucose is the most important source of energy in all organisms. An earlier work from the authors discovered that leucyl-tRNA synthetase 1 (LARS1) is controlled by the glucose availability to determine the metabolic fate of its cognate amino acid leucine (Yoon et al. Science 2020). In the presence of glucose, leucine is used for protein synthesis through the role of LARS1 both in charging the amino acid onto tRNA and in signaling mTOR activation. In the absence of glucose, LARS1 is phosphorylated by ULK1 at S391 and S720, which inhibits its leucine binding capacity and thus free the amino acid for oxidative energy production to aid cell survival. Remarkably, the current manuscript identified a key intermediate step between glucose starvation and LARS1 phosphorylation - O-GlcNAc modification at S1042 – and performed thorough analysis to understand its relationship to the downstream phosphorylation event (and its regulators such as AMPK and adolase) and its impact on leucine catabolism, ATP production, and cell protection. It is also commendable that the authors took the responsibility to compare LARS1 with Setrin1/2, the two identified leucine sensors for regulating mTOR/Rag at least in one of their experiments. The study is rigorous for the most part and deserves to be published in Nature Communications. However, we do have some comments that should be addressed before the publication.

1. A major concern we have is the lack of understanding on the O-GlcNAclation level of the purified LARS1 proteins used for in vitro studies (e.g., leucylation assay, circular dichroism, limited proteolysis). The LARS1 proteins (WT and S1042A mutant) were overexpressed and purified from in E. coli with or without exogenous OGT1 expression to represent WT LARS1 with or without O-GlcNAclation and the S1042A mutant (presumably without O-GlcNAclation in any case). However, it is likely that the modification is incomplete even when WT LARS1 is co-expressed with OGT1 in E. coli. Therefore, it is important to determine the stoichiometry of modified and unmodified populations in each purified protein sample and take it into consideration for interpreting the data from the in vitro studies.

2. As clearly shown in the model and cartoon in Extended Data Fig.8 and supported by data (e.g., knockdown of ULK1/2 activates mTOR (as indicated by p-S6K T389) even when LARS1 is O-GlcNAclated (Fig. 5e)), O-GlcNAc modification is required for ULK1 to phosphorylate LARS1, which blocks LARS1/RagD interaction, and O-GlcNAc modification itself cannot block LARS1/RagD interaction and inhibit mTOR. Therefore, it seems to be pointless to understand the conformational change induced by S1042 O-GlcNAclation for its impact on LARS1/RagD interaction (Fig. 4 a,b,c,d and Extended data Fig. 4). The impact of O-GlcNAclation should be on the LARS1/ULK1 interaction. What is the ULK1 binding site on LARS1 and its relationship to S1042 O-GlcNAclation?

3. For reasons explained above (#2), the statement in Abstract “Glucose starvation results in O-GlcNAcylation of LARS1 on residue S1042. This modification inhibits the interaction of LARS1 with RagD GTPase, so decreasing mTORC1 activity” is misleading.

4. In addition to S720, S391 site was also found to be phosphorylated by ULK1 in the Yoon et al. Science 2020 paper. Is S1042 O-GlcNAcylation also required for S391 phosphorylation?

5. It is shown in Fig. 3 a,b that the S6K T389 phosphorylation was also faster and more pronounced in cells expressing S1042 mutant than in cells expressing WT LARS1 when cultured with glucose (+Glu). This seems to suggest that the O-GlcNAcylation of LARS1 also occurs under normal conditions without glucose starvation. However, as shown in Fig.3c, a low O-GlcNAcylation level is observed for both WT and mutant cells under normal glucose condition and there is little difference in mTOR activation between WT and mutant cells. Thus, these 2 pieces of data seemed not consistent with each other.

Reviewer #2 (Remarks to the Author):

The authors recently reported an important discovery that LARS1 coordinates leucine-sensing and leucine catabolism with glucose availability, through LARS1 phosphorylation by ULK1. In the current manuscript, the authors reveal that LARS1 is O-GlcNAcyated on a single amino acid, S1042, in response to glucose starvation. They show that S1042 O-GlcNAcylation is a prerequisite for ULK1 phosphorylation of LARS1 on S720 and subsequent inhibition of mTORC1 activity via RagD. They further demonstrate that this mechanism is responsible for leucine-derived ATP production and protection from cell death upon glucose starvation. This is a well-designed and mechanistically very thorough study, and the manuscript is well written to present the findings in a logical fashion. The conclusions are supported by extensive and beautiful data. The central conclusion that LARS1 integrates the signals of amino acid availability and glucose availability is highly significant and impactful.

The vast majority of the western blots in the figures appear to be from single experiments. Authors mention quantification “by densitometry analysis of protein bands” in Methods, but no quantification data are shown for most of the western blots. I am not a fan of densitometry quantification of blots. However, given that the conclusions of this paper rest heavily on these western blots, the authors

should at least state in the figure legends whether independent experiments were performed with similar outcome and what the n was.

Reviewer #3 (Remarks to the Author):

The manuscript suggests that glucose availability regulates the central nutrient effector mTORC1 through intracellular leucine

sensor leucyl-tRNA synthetase 1 (LARS1). Glucose starvation results in O-GlcNAcylation of LARS1 on residue S1042. This modification inhibits the interaction of LARS1 with RagD GTPase, decreasing mTORC1 activity. This reduces the affinity of LARS1 for leucine by promoting phosphorylation of its leucine-binding site by the autophagy-activating kinase ULK1. The manuscript suggests that LARS1 integrates leucine and glucose availability to regulate mTORC1 and the metabolic fate of leucine.

The manuscript carefully documents these claims. Other than writing issues, the manuscript is convincing.

Reviewer #1

Q1. A major concern we have is the lack of understanding on the O-GlcNAclation level of the purified LARS1 proteins used for *in vitro* studies (e.g., leucylation assay, circular dichroism, limited proteolysis). The LARS1 proteins (WT and S1042A mutant) were overexpressed and purified from in *E. coli* with or without exogenous OGT1 expression to represent WT LARS1 with or without O-GlcNAclation and the S1042A mutant (presumably without O-GlcNAclation in any case). However, it is likely that the modification is incomplete even when WT LARS1 is co-expressed with OGT1 in *E. coli*. Therefore, it is important to determine the stoichiometry of modified and unmodified populations in each purified protein sample and take it into consideration for interpreting the data from the *in vitro* studies.

Answer: To address the reviewer #1's concern, we determined the stoichiometry of O-GlcNAcylated and unmodified LARS1 using O-GlcNAc Enzymatic Labeling System (Invitrogen, C33368, Click-IT™) for the *in vitro* modification of O-GlcNAc modified proteins. Based on this result, we observed that about 95% of WT LARS1 was O-GlcNAcylated, but the S1042A mutant was not O-GlcNAcylated at all when LARS1 was co-expressed with OGT1 in *E.coli*. We added this result to Extended Data Fig. 3a.

Q2. As clearly shown in the model and cartoon in Extended Data Fig.8 and supported by data (e.g., knockdown of ULK1/2 activates mTOR (as indicated by p-S6K T389) even when LARS1 is O-GlcNAclated (Fig. 5e)), O-GlcNAc modification is required for ULK1 to phosphorylate LARS1, which blocks LARS1/RagD interaction, and O-GlcNAc modification itself cannot block LARS1/RagD interaction and inhibit mTOR. Therefore, it seems to be pointless to understand the conformational change induced by S1042 O-GlcNAclation for its impact on LARS1/RagD interaction (Fig. 4 a,b,c,d and Extended data Fig. 4). The impact of O-GlcNAclation should be on the LARS1/ULK1 interaction. What is the ULK1 binding site on LARS1 and its relationship to S1042 O-GlcNAclation?

Answer: To avoid confusion of Fig. 5e, we added the short exposed p-S6K T389 blot as well as the previous long exposed p-S6K T389 blot. Because mTORC1 activity (as indicated by p-S6K T389) was not fully rescued by SBI-0206965 or ULK1/2 knockdown, we still believe that O-GlcNAcylation itself has inhibitory effect on mTORC1. We slightly modified the working model in Extended Data Fig. 8. According to the results in Fig. 4g and 5c, O-GlcNAcylation of LARS1 inhibits RagD binding and promotes ULK1 binding *in vitro*. Therefore, the conformational change induced by S1042 O-GlcNAcylation has at least two different roles on the control of LARS1-RagD binding and ULK1 binding-mediated S720

phosphorylation. As reviewer #1 suggested, we also analyzed the ULK1 binding site of LARS1 and the role of O-GlcNAcylation on ULK1 binding. Based on this result, the VC domain of LARS1 is the competitive binding site for RagD and ULK1 and O-GlcNAcylation of S1042 of LARS1 is required for ULK1 binding. We added these results to Extended Data Fig. 5c.

Q3. For reasons explained above (#2), the statement in Abstract “Glucose starvation results in O-GlcNAcylation of LARS1 on residue S1042. This modification inhibits the interaction of LARS1 with RagD GTPase, so decreasing mTORC1 activity” is misleading.

Answer: As reviewer #1 commented, we corrected the statement on Abstract.

Q4. In addition to S720, S391 site was also found to be phosphorylated by ULK1 in the Yoon et al. Science 2020 paper. Is S1042 O-GlcNAcylation also required for S391 phosphorylation?

Answer: To address the reviewer #1’s concern, we combined S391A and S720A mutants with phosphoserine antibody to monitor S391 phosphorylated LARS1 because phospho-LARS1 S391 antibody is not available. As a result, S391 is phosphorylated by glucose deprivation and this is mediated by O-GlcNAcylation as below.

Q5. It is shown in Fig. 3 a,b that the S6K T389 phosphorylation was also faster and more pronounced in cells expressing S1042 mutant than in cells expressing WT LARS1 when cultured with glucose (+Glu). This seems to suggest that the O-GlcNAcylation of LARS1 also occurs under normal conditions without glucose starvation. However, as shown in Fig.3c, a low O-GlcNAcylation level is observed for both WT and mutant cells under normal glucose condition and there is little difference in mTOR activation between WT and mutant cells. Thus, these 2 pieces of data seemed not consistent with each other.

Answer: In Fig. 3c, leucine-induced S6K1 phosphorylation was evaluated 15 minutes after leucine supplementation. Therefore, in Fig. 3a and b, we newly monitored p-S6K T389 up to 15 minutes instead of 10 minutes after leucine supplementation and replaced Fig. 3a and b with new data.

Reviewer #2

Q1. The vast majority of the western blots in the figures appear to be from single experiments. Authors mention quantification “by densitometry analysis of protein bands” in Methods, but no quantification data are shown for most of the western blots. I am not a fan of densitometry quantification of blots. However, given that the conclusions of this paper rest heavily on these western blots, the authors should at least state in the figure legends whether independent experiments were performed with similar outcome and what the n was.

Answer: Following the reviewer #2's comment, we have now adapted all figure legends and included a statement about how many experiments were performed with a similar outcome for each figure.

REVIEWERS' COMMENTS

Reviewer #1 (Remarks to the Author):

The authors addressed Q1 to Q3 satisfactorily.

Q4: Based on the data showing in the rebuttal letter, does it make sense to add the point that O-GlcNAc modification of LeuRS is also required for S391 phosphorylation? This result is consistent with the speculation in the Yoon et al. 2020 Science paper, which suggested that “phosphorylation of S391 might be a prerequisite for that of S720”.

Q5. The inconsistency between Fig. 3a,b and Fig. 3c was addressed, but the explanation was not satisfying. The authors added the 12 min and 15 min data points in Fig. 3a,b, so that by looking at the two new data points only, Fig. 3a,b and Fig. 3c are now consistent. However, the authors did not address the question why S1042A in LeuRS (compared with WT LeuRS) can activate S6K T389 phosphorylation without glucose starvation during the first 10 mins of leucine supplementation.

Reviewer #1:

Q1: Based on the data showing in the rebuttal letter, does it make sense to add the point that O-GlcNAc modification of LeuRS is also required for S391 phosphorylation? This result is consistent with the speculation in the Yoon et al. 2020 Science paper, which suggested that “phosphorylation of S391 might be a prerequisite for that of S720”.

Answer: Following reviewer #1's suggestion, we considered whether to include this data in the figure. Although we obtained the consistent result about the requirement of O-GlcNAcylation for S391 phosphorylation of LARS1, it is not clear the role of S391 phosphorylation on leucine signaling to mTORC1 and further studies are needed. In addition, we think that excluding this result will help general readers to easily understand the flow of the paper logically without affecting the conclusion at all. We hope you understand our judgment.

Q2. The inconsistency between Fig. 3a,b and Fig. 3c was addressed, but the explanation was not satisfying. The authors added the 12 min and 15 min data points in Fig. 3a,b, so that by looking at the two new data points only, Fig. 3a,b and Fig. 3c are now consistent. However, the authors did not address the question why S1042A in LeuRS (compared with WT LeuRS) can activate S6K T389 phosphorylation without glucose starvation during the first 10 mins of leucine supplementation.

Answer: As reviewer #1 pointed out, in Fig. 3a and b, S6K T389 phosphorylation occurs more rapidly in LARS1 S1042A-transfected cells than in LARS1 WT-transfected cells without glucose starvation during the first 10 min re-supplementation of leucine. One of the possible explanations of this result seem to be that the conformational change of LARS1 after leucine binding occurs faster in S1042A than in WT, so that mTORC1 is activated faster. In our previous report (Cell Reports 35, 109031, 2021), leucine binding induced conformation change of LARS1 for RagD binding. Especially, the rotation of swing helix (aa 893-918) corresponding to the front part of the VC domain (aa 893-1062) is critical for leucine-induced conformational change of RagD-binding site (aa 948-1015) of VC domain. And, S1042 is close to swing helix in 3 dimensional structure of the VC domain of LARS1. Therefore, alanine mutation of S1042 may make the swing helix easier to rotate and facilitate RagD binding.